# Oligodendrocytic Na+-K+-Cl⁻ co-transporter 1 activity facilitates axonal conduction and restores plasticity in the adult mouse brain

Yoshihiko Yamazaki [1,3✉], Yoshifumi Abe[2,3], Satoshi Fujii[1] & Kenji F. Tanaka [2]

The juvenile brain presents plasticity. Oligodendrocytes are the myelinating cells of the central nervous system and myelination can be adaptive. Plasticity decreases from juvenile to adulthood. The mechanisms involving oligodendrocytes underlying plasticity are unclear. Here, we show Na+-K+-Cl⁻ co-transporter 1 (NKCC1), highly expressed in the juvenile mouse brain, regulates the oligodendrocyte activity from juvenile to adulthood in mice, as shown by optogenetic manipulation of oligodendrocytes. The reduced neuronal activity in adults was restored by *Nkcc1* overexpression in oligodendrocytes. Moreover, in adult mice over-expressing *Nkcc1*, long-term potentiation and learning were facilitated compared to age-matched controls. These findings demonstrate that NKCC1 plays a regulatory role in the age-dependent activity of oligodendrocytes, furthermore inducing activation of NKCC1 in oligo-dendrocytes can restore neuronal plasticity in the adult mouse brain.

[1] Department of Physiology, Yamagata University School of Medicine, Yamagata, Japan. [2] Department of Neuropsychiatry, Keio University School of Medicine, Tokyo, Japan. [3]These authors contributed equally: Yoshihiko Yamazaki, Yoshifumi Abe. ✉email: yyamazak@med.id.yamagata-u.ac.jp

Neural development is initiated in the embryonic period and myelination of axons is mostly completed in the juvenile period. In the rodent hippocampus, the neural circuits are formed by synaptogenesis, tuned by synapse maturation, and consolidated by myelination, resulting in high-speed conduction (Fig. 1a). Neural plasticity, including synaptic plasticity, is high during the juvenile period and declines in the fully myelinated adult brain (Fig. 1a); therefore, myelination appears to be a sign of the weakening of neural plasticity. However, myelination adapts throughout adulthood and contributes to maintain functional plasticity[1–4].

In the white matter, the density of oligodendrocytes (OLs) and the myelin they produce is high. Previously, we found a unique axon-OL interaction beyond saltatory conduction in the rodent hippocampus[3,5]. We reported that myelinating OLs were depolarized after neuronal excitation, which was regarded as a forward action from neurons to OLs. We depolarized OLs using electrophysiological or optogenetic methods and demonstrated that this modulation results in the short- and long-term facilitation of axonal conduction[3,5], which was regarded as a reverse action from OLs to neurons. It is possible that this neuron-OL interaction regulates myelinated fiber plasticity in adulthood. However, the age-dependent profiles of OL-related plasticity, its underlying molecular mechanism, and its influence on neural circuits remain to be elucidated.

OLs and myelin show morphological adaptive changes in response to neural activity, affecting axonal conduction[6]. The intracellular volume of the myelinating processes is small, neural activity-induced OL depolarization leads to an increase in the volume of myelinating processes as a result of osmotic swelling[7]. The application of GABA to OLs induces depolarizing responses[8] due to the presence of $Cl^-$ transporters[9]. $Na^+–K^+–Cl^-$ co-transporter 1 (NKCC1) is a regulator of cell volume. We hypothesize that oligodendrocytic NKCC1 expression may regulate myelinated fiber plasticity.

In the present study, we investigate whether myelinated fiber plasticity is also age-dependent, reconciling age-dependent synaptic plasticity[10,11] with adaptive myelination. We find that the magnitude of axonal plastic changes induced by optogenetic-mediated OL depolarization is larger in juvenile than adult mice and is related to *Nkcc1* expression and its activity in OLs. *Nkcc1* overexpression in adult mice or its knockdown in juvenile mice facilitates or suppresses the induction of myelinated fiber plasticity-related long-term potentiation (LTP), respectively. Hippocampal-dependent learning is enhanced in *Nkcc1*-over-expressing compared to age-matched control mice. Altogether, these results demonstrate that NKCC1 activity accounts for age-dependent myelinated fiber plasticity and that the rejuvenation of OLs mediated by NKCC1 facilitates neural function in the adult brain.

## Results

**OL depolarization-induced plasticity varies with development**. Immunohistochemistry with the myelin marker proteolipid protein (PLP) demonstrated that myelination was initiated at approximately postnatal day (PND) 10 and reached a peak at

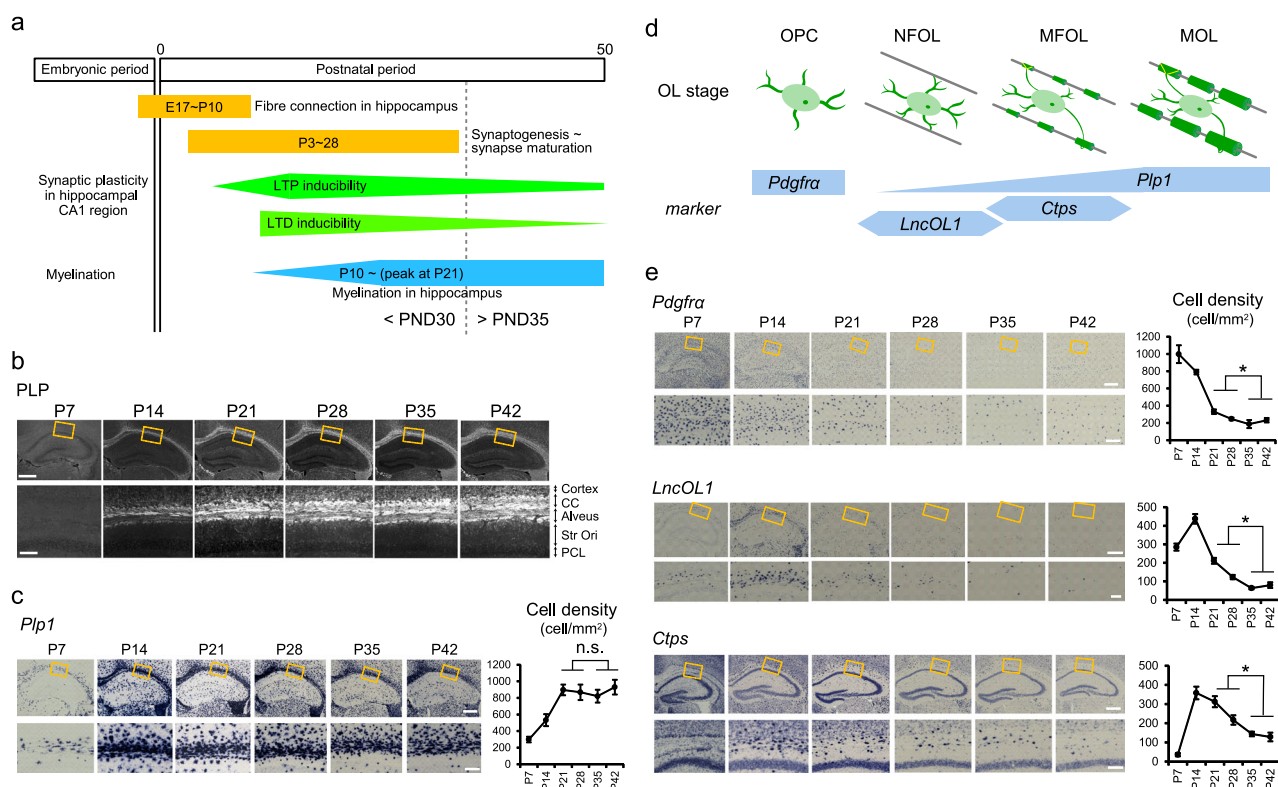

**Fig. 1 Developmental changes in myelination and gene expression of OL stage markers. a** Summary of known developmental changes in neural circuits, inducibility of synaptic plasticity, and myelination in rodents. **b**, **c** Developmental changes in immunoreactivity for proteolipid protein (PLP) (**b**) and gene expression of *Plp1* (**c**) in the alveus of the hippocampus. Data are presented as mean ± SEM. **d** Schematic drawing showing OL stages and their markers. **e** Developmental changes in the expression of *Pdgfrα*, *LncOL1*, and *Ctps* mRNA in the hippocampus. The graphs to the right of each in situ hybridization image indicate the quantitative analysis data using 8 slices from 2 animals of each gene for the indicated postnatal day. Cells are counted in the area including the corpus callosum, alveus and stratum oriens. Unpaired two-sided Student's *t* test, *Pdgfrα*, $t_{30} = 2.96$, $P = 0.0062$; *LncOL1*, $t_{30} = 6.28$, $P < 0.001$; *Ctps*, $t_{30} = 5.96$, $P < 0.001$. Scale bar, 500 and 100 μm for lower and higher magnification images, respectively. Data are presented as mean ± SEM. *$P < 0.05$. Source data are provided as a Source Data file.

PND 21 in the white matter of the mouse hippocampus (Fig. 1b). *Plp1* levels were maintained in myelinated fibers after PND 21, implying full white matter maturation at this stage (Fig. 1c). A recent single-cell RNA-sequencing study reported that OLs can be classified by their developmental stage: newly formed OLs (*LncOL1*- and *Plp1*-positive), myelin-forming OLs (*Ctps*- and *Plp1*-positive), and mature OLs (*Plp1*-positive and *Ctps*-negative) (Fig. 1d, Supplementary Fig. 1)[12]. To reconcile this single-cell RNA-sequencing-based classification with the conventional classification system (premyelinating OLs and mature OLs), we conducted double labeling with DM20, an isoform of PLP protein and a marker of premyelinating OLs[13], and respective mRNAs. DM20 was expressed by *LncOL1*-positive cells (Supplementary Fig. 2a), indicating that newly formed OLs are equivalent to premyelinating OLs.

The number of *Plp1*-positive OLs and level of PLP immunoreactivity were comparable before PND 30 and after PND 35 (Fig. 1b, c); however, there were more young OLs (newly formed and myelin-forming OLs) before PND 30 than after PND 35 (Fig. 1e). That is, myelin staining or *Plp1* mRNA labeling does not distinguish between young (<PND 30) and old (>PND 35) myelinated fibers. However, at younger time points, the fibers are myelinated by more young OLs.

We previously developed a method to assess OL-dependent neural plasticity. In this method, OLs expressed channelrhodopsin-2 (ChR2) under the control of the *Plp1* promoter (PLP-ChR2) (Fig. 2a)[3], and OL precursor cells (NG2-positive) and DM20-positive premyelinating OLs (newly formed OLs) did not express ChR2 (Fig. 2b, Supplementary Fig. 2b), indicating that the OL precursor cells/premyelinating OLs were not optogenetically manipulated, while myelin-forming and mature OLs were manipulated. ChR2 expression did not alter the number of OL lineage cells at PND 21 and 42 (Supplementary Fig. 3a), consistent with previous data regarding OL differentiation[12] (Supplementary Fig. 1). Optical stimulation mimicked the physiological relevant depolarization of OLs, i.e., the magnitude of depolarization by ChR2 activation was comparable to that induced by theta rhythm electrical stimulation, which reflects physiological activity in the hippocampus[3]. As a result, ChR2-mediated OL depolarization induced short- and long-term plastic changes in axonal conduction, that is, increased conduction velocity and enhanced axonal excitability, respectively[3,4]. By exploiting this optogenetic approach, we examined whether OL-dependent plastic changes were age-dependent. According to the developmental changes in myelination and gene expression of OL stage markers, we divided the PLP-ChR2 mice into younger (≤PND 30) and older (>PND 35) groups.

To probe axonal conduction, we recorded antidromic action potentials by whole-cell recording from CA1 pyramidal cells with stimulation of their axons and compound action potentials (CAPs) by extracellular recording in the alveus (Fig. 2c, d). We calculated the mean latency of antidromic action potentials and the mean amplitude of CAPs at 1–3 and 28–30 min after photostimulation, respectively, to evaluate the plastic changes in axonal conduction induced by OL depolarization. As we demonstrated previously[3,4], the latency of action potentials was transiently decreased in PLP-ChR2 mice (Fig. 2c, Supplementary Fig. 4a), reflecting increased conduction velocity in the axons of CA1 pyramidal cells, and the increased CAP amplitude was sustained (Fig. 2d, Supplementary Fig. 4b), reflecting enhanced CA1 axonal excitation. Of note, younger mice showed more plastic changes in both parameters (latency of action potential: $t_{16} = 3.23$, $P = 0.0052$; CAP amplitude: $t_{20} = 2.18$, $P = 0.042$ vs >PND 35) (Fig. 2c, d, Supplementary Fig. 4a, b). We validated the increase in conduction velocity and its conspicuousness in younger mice in a more rigorous way by evaluating the changes in latency differences ($t_{13} = 2.42$, $P = 0.030$ vs >PND 35)

(Supplementary Fig. 5). We confirmed that the magnitude of ChR2-mediated OL depolarization at ≤ PND 30 ($n = 3$) was comparable to that at >PND 35 ($n = 4$) (peak: $F_{(1,40)} = 2.23$, $P = 0.16$; area: $F_{(1,40)} = 1.34$, $P = 0.31$, two-way repeated-measures ANOVA; Supplementary Fig. 4c). These data indicated that OL-mediated axonal plasticity was age-dependent.

OL depolarization is triggered by neural activation in physiological conditions. Besides glutamate, ATP, and extracellular $K^+$, GABA can also induce OL depolarization[8], due to the accumulation of $Cl^-$ in OLs by active $Cl^-$ transport[9]. Therefore, we examined NKCC1 expression. *Nkcc1* mRNA was predominantly expressed in the white matter of younger mice (≤PND 30), and it was expressed at a low level in older mice (>PND 35) (Fig. 2e). Its temporal expression pattern was similar to that of the young OL markers *LncOL1* and *Ctps* (Fig. 1e). Double fluorescence in situ hybridization showed the co-expression of *Nkcc1* and *Plp1*, *Ctps*, or *LncOL1* mRNA, but not *Pdgfrα* mRNA, at PND 21 (Fig. 2f), which was consistent with previous single cell RNA-sequencing data (Supplementary Fig. 1). These data indicated that *Nkcc1* was expressed in *Plp1*-, *Ctps*-, or *LncOL1*-positive OLs and its expression level was age-dependent.

We next examined whether NKCC1 expression in OLs was functional. We applied GABA (1 mM) to the recorded OLs and measured the amplitude of GABA-induced inward currents (Fig. 2g). We also calculated the bumetanide-sensitive component of inward currents (Fig. 2h, i); bumetanide inhibits NKCC1 activity and reduces $Cl^-$ accumulation, leading to the suppression of GABA-induced $Cl^-$ efflux[14]. The identification of the cells as OLs was confirmed by *post hoc* staining with injected biocytin (Fig. 2g). The resting membrane potential and input resistance of OLs were comparable between younger mice (≤PND 30) and older mice (>PND 35) ($t_{42} = 0.48$, $P = 0.64$ for resting membrane potential and $t_{42} = 0.16$, $P = 0.88$ for input resistance) (Supplementary Fig. 6a). Similar to *Nkcc1* mRNA levels (Fig. 2e), the amplitude and bumetanide-sensitive component of GABA-induced inward currents were age-dependent (Fig. 2i, Supplementary Fig. 6b).

We examined whether higher *Nkcc1* levels were responsible for the increased axonal plasticity driven by ChR2-mediated OL depolarization in younger mice. We compared the degree of increased conduction velocity and enhanced axonal excitation in the presence or absence of bumetanide. We found that NKCC1 blockade abolished the additive effects in younger mice (latency of action potential: $t_{14} = 3.96$, $P = 0.0014$; CAP amplitude: $t_{20} = 2.55$, $P = 0.019$ vs the absence of bumetanide) (Fig. 2c, d, Supplementary Fig. 4a, b). We confirmed that the number of *Nkcc1*-expressing cells and their age-dependent decline were not affected by ChR2 expression on OLs (Supplementary Fig. 3b). These results suggested that the plasticity of myelinated fibers revealed by ChR2-manipulating methods was age-dependent and that this age-dependent facilitation coincided with high levels of *Nkcc1* in OLs.

**OL depolarization-induced plasticity depends on its distance from the neuron soma.** Regarding the short-term plastic changes, we found a different aspect to the effect of OL depolarization on axonal conduction velocity, i.e., the magnitude of this effect varied according to its distance from the neuron soma. We recorded the antidromic action potentials of various latencies by stimulating the axons at different positions and examined the effect of OL depolarization on axonal velocity (Fig. 3a, b). We first performed these experiments in older PLP-ChR2 mice (>PND 35). Figure 3c shows the relationship, measured in 92 axons (each shown as a gray point), between the latency of the action potential before photostimulation and the magnitude of the change in

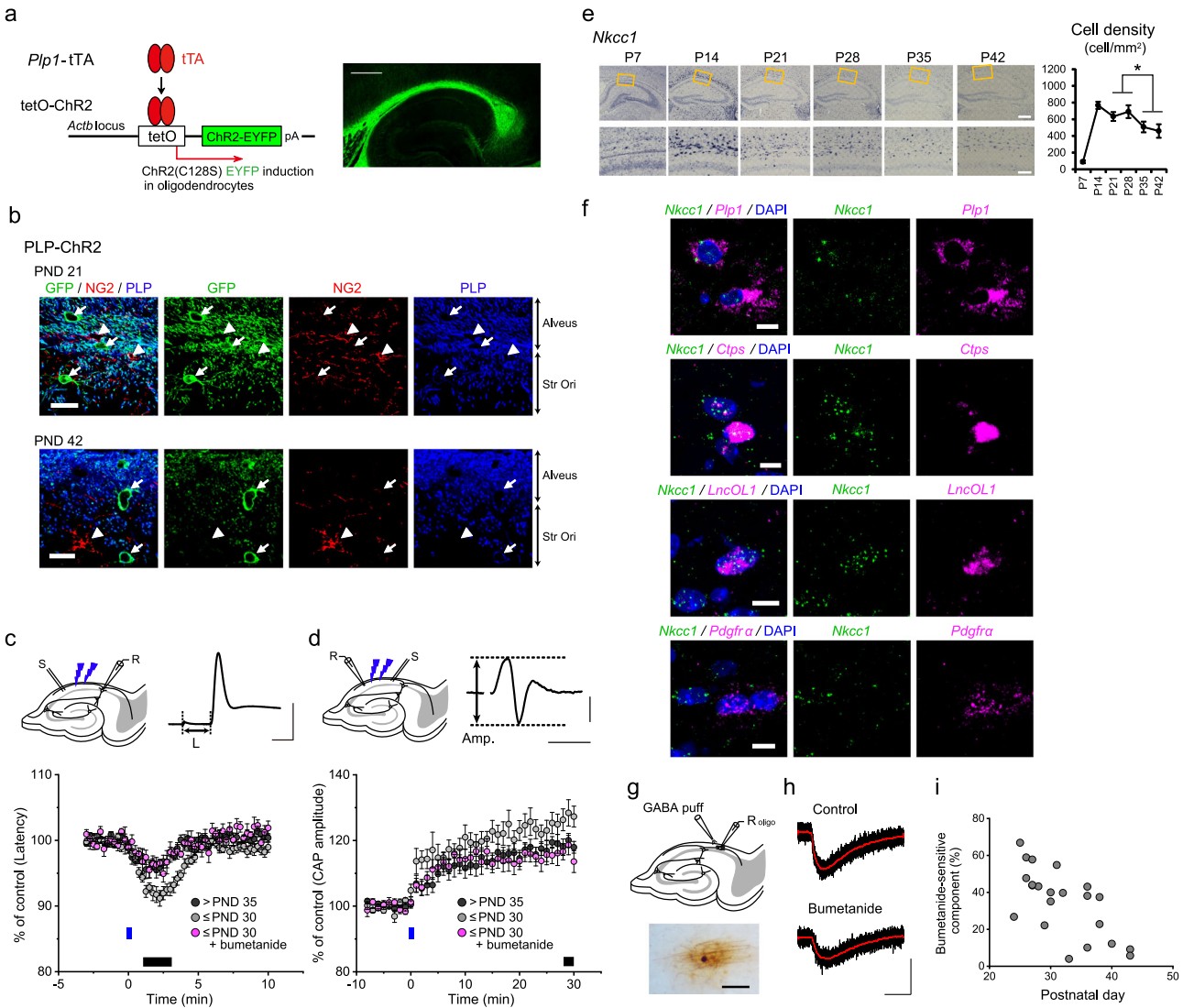

**Fig. 2 OL-mediated plasticity of axonal conduction and NKCC1 activity vary with postnatal development. a** Induction of OL-specific channelrhodopsin-2 (ChR2) (C128S)-EYFP expression in the tetracycline system and fluorescent image of the hippocampus labeled with an anti-GFP antibody. Scale bar, 500 μm. **b** Immunohistochemistry showing that EYFP-expressing cells are PLP-positive (arrow), but NG2-negative (arrowhead), in PLP-ChR2 mice at PND 21 and 42. Scale bars, 10 μm. Similar results were observed in three mice. **c** Recording of antidromic action potentials in CA1 pyramidal cells and measurement of conduction latency (L). Scale, 2 ms, 50 mV. Time-course of the latency of action potentials after photostimulation in PLP-ChR2 mice at > PND 35 ($n = 9$, PND 39–58), and ≤ PND 30 in the absence ($n = 9$, PND 18–26) or presence ($n = 7$, PND 20–30) of bumetanide. Data are presented as mean ± SEM. **d** Compound action potential (CAP) recording in the alveus and measurement of CAP amplitude (Amp.). Scale, 5 ms, 0.5 mV. Changes in CAP amplitude induced by blue light photostimulation in PLP-ChR2 mice > PND 35 ($n = 10$, PND 38–57) and ≤ PND 30 in the absence ($n = 12$, PND 21–28) or presence ($n = 10$, PND 19–28) of bumetanide. Data are presented as mean ± SEM. Black bars in the graphs of (**c**) and (**d**) indicate the measurement periods in Supplementary Fig. 4b, c. **e** In situ hybridization (ISH) images with an Nkcc1 mRNA probe showing signals in the hippocampus (top panels) and alveus (bottom panels) in wild-type mice. Scale bar, 500 μm (top panels) and 100 μm (bottom panels). Quantitative analysis data of Nkcc1 ISH using 8 slices from 2 animals. Unpaired two-sided Student's t test, $t_{30} = 3.30$, $P = 0.0025$. Data are presented as mean ± SEM. *$P < 0.05$. **f** Fluorescence ISH with Nkcc1/Plp1, Nkcc1/Ctps, Nkcc1/LncOL1, and Nkcc1/Pdgfrα mRNA probes in the alveus of wild-type mice at PND 21. Scale bar, 20 μm. Similar results were observed in three mice. **g** Schematic drawing showing the recording (R oligo) and GABA puffer pipettes, and a typical example of a recorded OL stained with injected biocytin. Scale bar, 50 μm. **h** GABA-induced currents in the absence and presence of bumetanide. Scale, 2 s and 50 pA. **i** Relationship between PND and the bumetanide-sensitive component of GABA-induced inward currents ($n = 22$). PND in this and following figures indicates postnatal day. Source data are provided as a Source Data file.

latency induced by OL stimulation (1–3 min after photostimulation, expressed as % change), while Fig. 3d shows the summarized results binned with every 0.25 ms of latency. At a short latency position (0.75–1.0 ms, mean distance from the soma: $187.2 ± 40.7$ μm [$n = 10$]), the change in latency was not large, but began to increase gradually at a latency of 1.25 ms. At the ~2 ms latency position, the effect of depolarization on velocity was significant, with a > 5% change at a latency of 1.5–2.5 ms and

an $8.2 ± 1.3\%$ change at a latency of 1.75–2.0 ms (mean distance from the soma: $722.9 ± 64.4$ μm [$n = 7$]) (Fig. 3e). Although significant changes were observed at even longer latency positions, the magnitude was small compared to those seen at the ~2.0 ms latency position. These data suggest the presence of a region in which the OL depolarization-induced effect was more evident than in the other regions (Fig. 3f). Compared with older mice, younger mice (≤PND 30) retained the effect of depolarization on

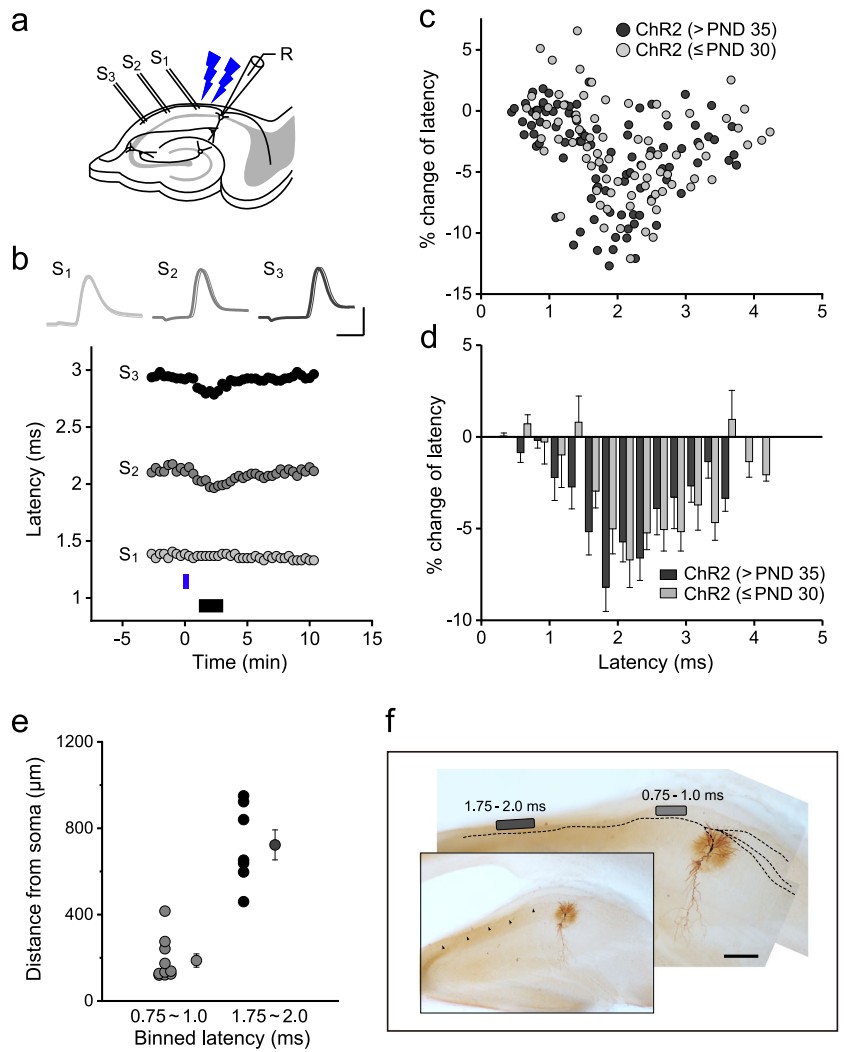

**Fig. 3 OL depolarization-induced increase in conduction velocity varies according to axonal position. a** Schematic drawing for recording antidromic action potentials with different stimulating positions ($S_1$–$S_3$) in a CA1 pyramidal cell. **b** Time course of the latency of action potentials evoked at different positions after photostimulation. The sample traces in the inserts indicated by a thin or thick line were recorded before and at 2 min after photostimulation, respectively. Scale, 2 ms and 50 mV. **c, d** Relationship between the latency before OL depolarization and the change in latency induced by OL depolarization in PLP-ChR2 mice. Distribution of the results for the percentage change in latency for separate experiments ($n = 92$ for > PND 35 [42–62] and $n = 77$ for ≤ PND 30 [18–26]) (**c**) and the summarized data in 0.25-ms bins (**d**). Data are presented as mean ± SEM. **e** Distance from the stimulating electrode to the soma of the recorded cells along the axon for action potentials with a latency of 0.75–1.0 or 1.75–2.0 ms. **f** Example of axonal position showing the most evident change in latency (dark gray bar) and no significant change in latency (light gray bar). Scale, 100 μm. Source data are provided as a Source Data file.

velocity even at the distant electrode position ($n = 77$, >5% change at 1.75–3.0 ms latency) (Fig. 3c, d), suggesting the extension of the zone exhibiting a significant effect in the juvenile period. These results provided additional data supporting the age-dependent plasticity of myelinated fibers.

***Nkcc1* overexpression restores OL depolarization-induced neuronal plasticity.** As shown above, myelinated fiber plasticity was high in the juvenile brain. Pharmacological NKCC1 loss-of-function studies have claimed that NKCC1 activity is a decisive factor for age-dependent neuronal plasticity. We sought to assess whether sustained *Nkcc1* gene expression in OLs in older mice could restore myelinated fiber plasticity to the level observed in younger mice. To this end, we generated OL-specific *Nkcc1*-overexpressing and ChR2-expressing mice (Fig. 4a) and examined plasticity using our optogenetic approach.

We crossed the *Nkcc1*[tetO] knock-in line[15] with the *Plp1*-tTA line to establish tTA-dependent *Nkcc1* induction. In situ hybridization showed the overexpression of *Nkcc1* mRNA in adult mice (>PND 42) compared with age-matched control mice (*Nkcc1*[tetO/+]) (Fig. 4a). *Plp1*-, *Ctps*-, and *LncOL1*-positive cells strongly expressed *Nkcc1* mRNA, but *Pdgfrα*-positive cells did not, indicating *Nkcc1* mRNA overexpression after newly-formed OL stages (Fig. 4b). The number of OL lineage cells in *Nkcc1*-overexpressing mice at PND 42 was comparable to that in control mice (Supplementary Fig. 7a). Axon diameter, myelin thickness, g-ratio, and myelinated axon density were comparable between *Nkcc1*-overexpressing mice and control mice at PND 42 (myelin thickness: $t_6 = 0.24$, $P = 0.82$; axon diameter: $t_6 = 0.66$, $P = 0.54$; g-ratio: $t_6 = 0.78$, $P = 0.46$; density: $t_6 = 2.76$, $P = 0.064$, Supplementary Fig. 7b–d). These data indicated that *Nkcc1* overexpression did not affect OL development. The resting membrane potentials and input resistances of OLs were not significantly

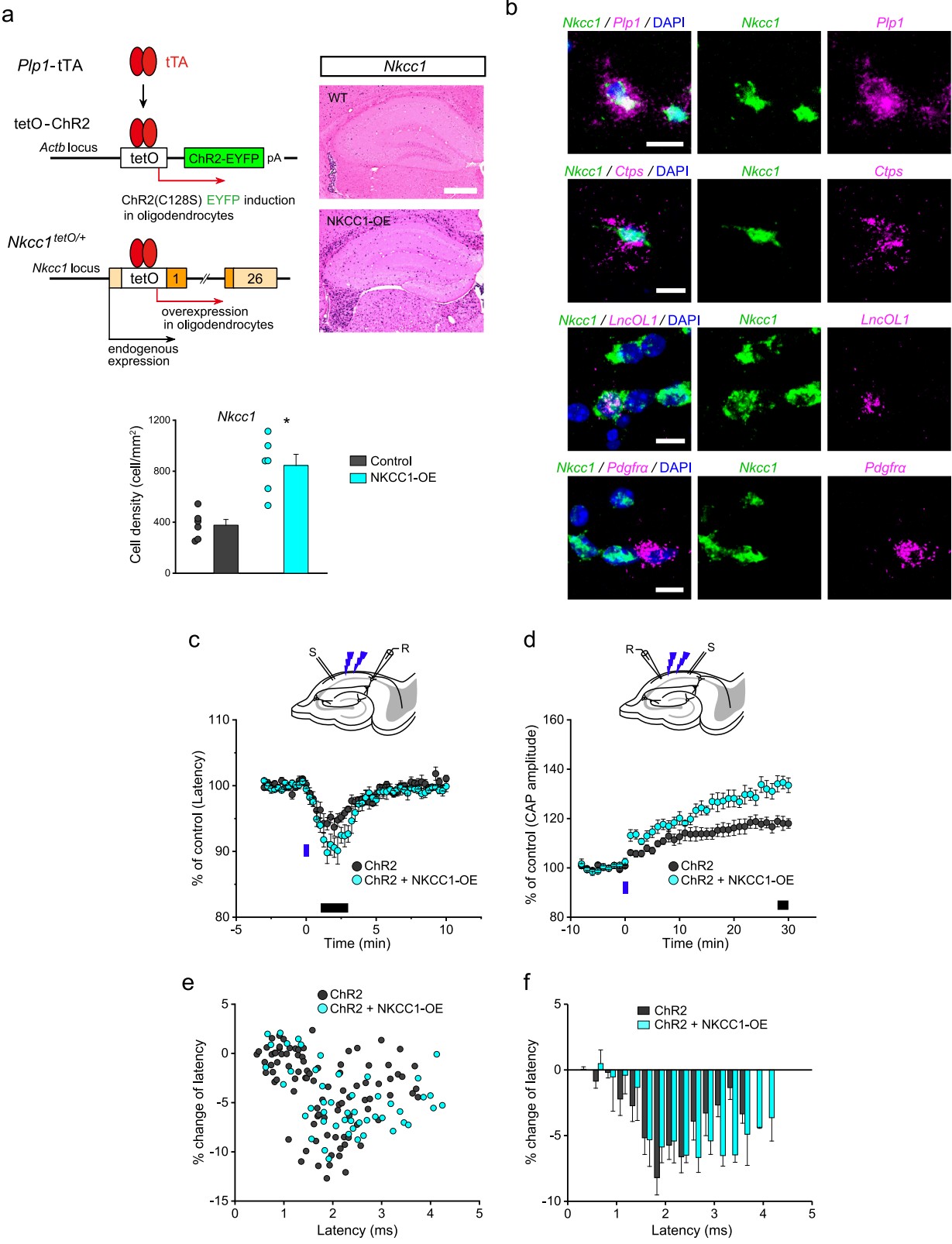

different between *Nkcc1*-overexpressing and control mice ($t_{36} = 0.82$, $P = 0.42$ for resting membrane potential and $t_{36} = 1.41$, $P = 0.17$ for input resistance) (Supplementary Fig. 8a), and internodal length was comparable between both groups (Supplementary Fig. 8b), indicating that *Nkcc1* overexpression did not affect basal cellular properties.

The function of *Nkcc1* overexpressed on OLs (>PND 35) was confirmed by the larger magnitude of GABA-induced inward

**Fig. 4 Nkcc1 overexpression in OLs facilitates OL-mediated plasticity of axonal conduction. a** Induction of OL-specific co-expression of channelrhodopsin-2 (ChR2) (C128S)-EYFP and *Nkcc1*. Hippocampal in situ hybridization images of the *Nkcc1* mRNA probe in wild-type (WT) and *Nkcc1*-overexpressing (NKCC1-OE) mice at PND 55. Scale bar, 500 μm. Cell counts for *Nkcc1*-positive cells in the area including the corpus callosum, alveus, and stratum oriens in control mice (*Nkcc1*tetO/+) and NKCC1-OE mice at PND 42 (*n* = 6 slices from 3 animals). Data are presented as mean ± SEM. *P < 0.05. **b** Fluorescence in situ hybridization images with *Nkcc1/Plp1*, *Nkcc1/Ctps*, *Nkcc1/LncOL1*, and *Nkcc1/Pdgfrα* mRNA probes detected in the alveus in NKCC1-OE mice at PND 42. Scale bar, 10 μm. **c** Recording of antidromic action potentials in a CA1 pyramidal cell. Time-course of the latency of action potentials after photostimulation in PLP-ChR2 mice (*n* = 12, PND 37–58) and PLP-ChR2 mice overexpressing *Nkcc1* (*n* = 8, PND 37–56). Data are presented as mean ± SEM. **d** Recording of compound action potentials (CAPs) in the alveus. Changes in CAP amplitude induced by blue light photostimulation in PLP-ChR2 mice (*n* = 10, PND 39–52) and PLP-ChR2 mice overexpressing *Nkcc1* (*n* = 8, PND 38–60). Data are presented as mean ± SEM. Black bars in the graphs of (**c**) and (**d**) indicate the measurement periods in Supplementary Fig. 4b, c. **e, f** Relationship between the latency before OL depolarization and the change in latency induced by OL depolarization in PLP-ChR2 mice overexpressing *Nkcc1*. Distribution of the results for the percentage change in latency (*n* = 49, light blue dots) (**e**) and the summarized data (light blue columns, 0.25-ms bins) (**f**). Data are presented as mean ± SEM. The dark gray dots and columns correspond to the data of >PND 35 mice shown in Fig. 3c, d. Source data are provided as a Source Data file.

currents ($t_7 = 4.81$, $P = 0.0019$ vs ChR2 expression alone) and bumetanide-sensitive component of the currents ($t_7 = 4.73$, $P = 0.0021$ vs ChR2 expression alone) (Supplementary Fig. 8c, d). The magnitude of ChR2-mediated OL depolarization was comparable between groups (peak: $F_{(1,48)} = 3.16$, $P = 0.11$; area: $F_{(1,48)} = 1.50$, $P = 0.23$, two-way repeated-measures ANOVA; Supplementary Fig. 9a). Using these *Nkcc1*-overexpressing adult mice (>PND 35), we examined ChR2-mediated OL depolarization induced short- and long-term plastic changes. Increased axonal velocity and enhanced axonal excitability were facilitated in these mice (latency of action potential: $t_{18} = 2.11$, $P = 0.048$; CAP amplitude: $t_{16} = 3.65$, $P = 0.0022$ vs ChR2 expression alone) (Fig. 4c, d, Supplementary Fig. 9b, c), demonstrating that *Nkcc1* overexpression in adult mice restored plasticity to a similar level as observed in the juvenile brain.

We further assessed whether the axonal active zone for axonal plasticity was affected by *Nkcc1* overexpression in adult mice. In older PLP-ChR2 mice overexpressing *Nkcc1* (>PND 35), the region exhibiting the effects of OL depolarization was larger than that in older PLP-ChR2 mice (>5% change in 1.5–3.5 ms latency) (Fig. 4e, f). The extension of the active zone was similar to that seen in juvenile mice, supporting the view that *Nkcc1* overexpression in OLs restored the effects of depolarization on conduction velocity.

**Manipulation of oligodendrocyte *Nkcc1* expression affects the induction of LTP.** We previously demonstrated that the myelinated fiber plasticity promotes synaptic plasticity at destination synapses between myelinated axons and the target postsynaptic neurons[4]. Given the presence of age-dependent axonal plasticity, the higher synaptic plasticity in the juvenile period should be affected by impairing axonal plasticity, and the lower synaptic plasticity in the adult period should be improved by facilitating axonal plasticity. To prove this, we examined CA1-subiculum synaptic plasticity in *Nkcc1* loss-of-function juvenile mice and *Nkcc1* gain-of-function adult mice. Since the modulatory effects of OL depolarization on the responses of destination synapses were apparent in bursting pyramidal cells located in the mid or distal region of the subiculum[4], excitatory postsynaptic currents (EPSCs) were recorded in these cells (Fig. 5a). EPSCs were evoked by electrical stimulation to the axons of CA1 pyramidal cells and their amplitudes were measured to analyze the changes in EPSCs (Fig. 5a). Since the effects of OL depolarization on axonal conduction differed according to axonal position (Fig. 3), it is possible that the effects on synaptic function at destination synapses vary depending on the site of stimulation. To minimize this possible influence, we made the linear distance between the stimulating and recording electrodes constant (approximately 600 μm).

We generated doxycycline (Dox)-controllable *Nkcc1* knockdown mice (*Actin*-tTS::*Nkcc1*tetO/tetO)[15] and control mice (*Nkcc1*tetO/tetO), and used juvenile and adult mice (≤PND 30 and PND = 42, respectively) (Fig. 5b). In situ hybridization showed the ubiquitous knockdown of *Nkcc1* mRNA in the hippocampus at PND 21 and 42 compared with age-matched control mice (Fig. 5c). The number of OL lineage cells was comparable between *Nkcc1* knockdown mice and control mice at PND 21 and 42 (Supplementary Fig. 10a), except for the number of *LncOL1*-positive cells at PND 21. In super-resolution microscopy analysis, axon diameter, myelin thickness, and g-ratio were comparable between *Nkcc1* knockdown mice and control mice at PND 21 and 42 (axon diameter: $t_6 = 0.66$, $P = 0.54$; myelin thickness: $t_6 = 0.24$, $P = 0.82$; g-ratio: $t_6 = 0.78$, $P = 0.46$) (Supplementary Fig. 10b–d). *Nkcc1* knockdown mice showed a low density of myelinated fibers at PND 21. The membrane properties of OLs were not different between both genotypes ($t_{27} = 0.50$, $P = 0.62$ for resting membrane potential and $t_{27} = 0.44$, $P = 0.67$ for input resistance) (Supplementary Fig. 11a). The postnatal knockdown of *Nkcc1* expression in OLs was confirmed by the lowered magnitude and bumetanide-sensitive component of GABA-induced currents in OLs (magnitude: $t_{12} = 3.56$, $P = 0.0039$; bumetanide-sensitive component: $t_{12} = 7.77$, $P < 0.001$) (Supplementary Fig. 11b, c).

We then examined whether *Nkcc1* knockdown in OLs affected the induction of LTP in CA1-subiculum synapses. We applied theta burst stimulation consisting of a different number of bursts (10, 15, or 20 bursts) with each burst containing 4 pulses at 100 Hz and individual bursts separated by 200 ms (Fig. 5a). In young control mice, 15 and 20 bursts, but not 10 bursts, induced a significant increase in the amplitude of EPSCs at 35–40 min after burst stimulation (15 bursts: $t_5 = 5.07$, $P = 0.0039$; 20 bursts: $t_7 = 5.71$, $P < 0.001$ vs baseline) (Fig. 5d, e), suggesting that the threshold of LTP induction in younger animals was between 10 and 15 bursts, which was lower than in adult mice[4]. In *Nkcc1* knockdown mice, the magnitude of LTP was suppressed (two-way ANOVA: $F_{(1,41)} = 9.80$, $P = 0.0032$ vs control mice) (Fig. 5d, e). These results indicated that the *Nkcc1* loss-of-function-mediated lowering of axonal plasticity disturbed the readily inducible synaptic LTP in juvenile mice. We next examined whether *Nkcc1* overexpression in OLs (Fig. 6a) facilitated the induction of LTP in adult mice. In wild-type (WT) control mice (>PND 35), LTP was induced by the application of 20 bursts ($t_5 = 3.68$, $P = 0.0014$), but not by 5, 10, or 15 bursts (Fig. 6c, d), indicating that the threshold for LTP induction was between 15 and 20 bursts, which was a higher than in juvenile mice. In OL *Nkcc1*-overexpressing mice, LTP induction was enhanced (two-way ANOVA: $F_{(1,47)} = 7.17$, $P = 0.010$ vs WT mice) and the threshold for LTP induction was less than 10 bursts, with a larger increase in EPSCs than in WT mice (10 bursts: $t_9 = 2.41$, $P = 0.043$; 15 bursts, $t_{13} = 3.05$,

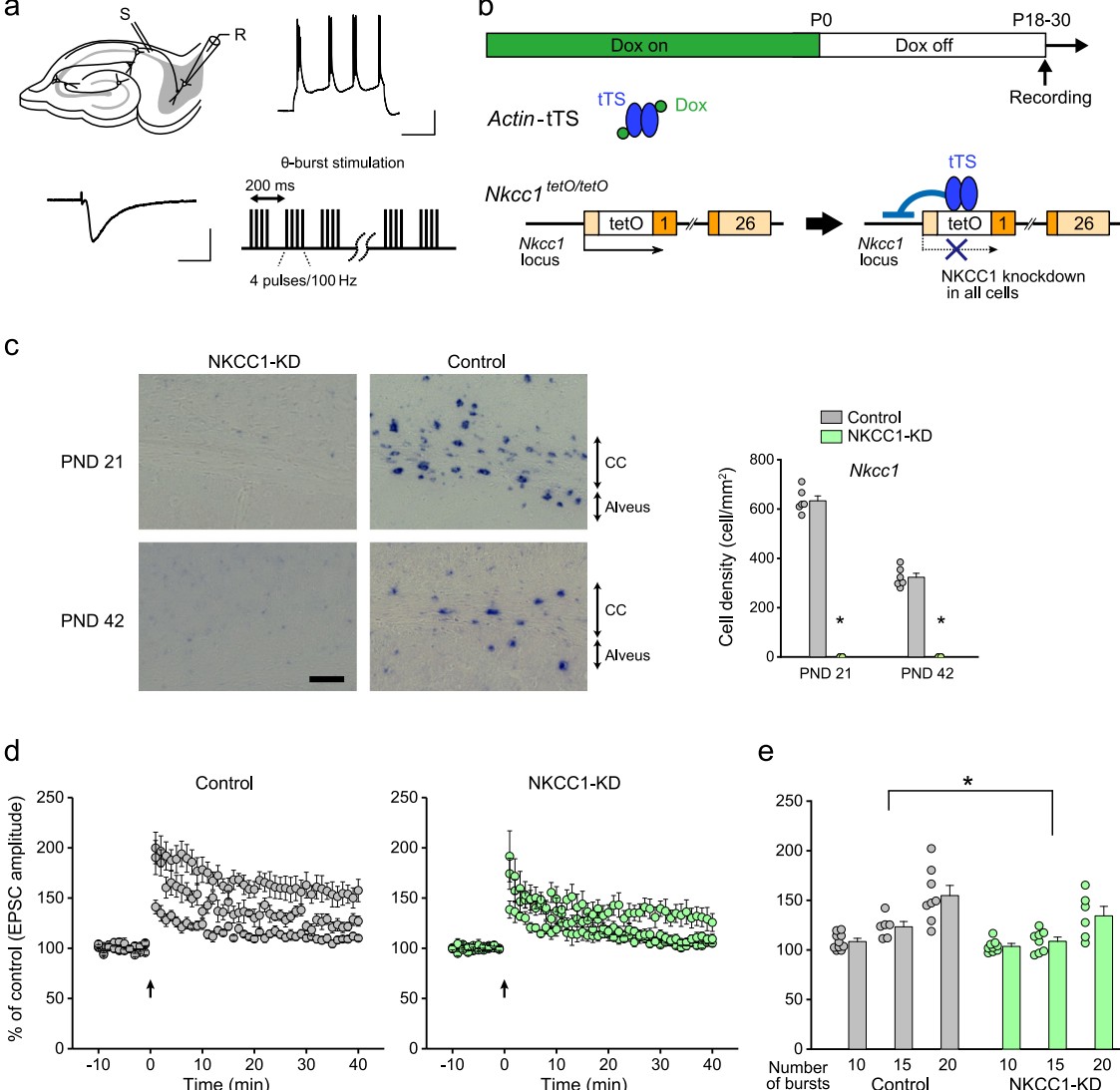

**Fig. 5 Nkcc1 loss-of-function attenuates the induction of long-term potentiation. a** Recording of excitatory postsynaptic currents (EPSCs) from bursting neurons in the subiculum and electrical stimulation protocol of theta burst stimulation for the induction of long-term potentiation. Scale, 200 ms and 20 mV for firing pattern, and 50 ms and 100 pA for EPSC. **b** Induction of *Nkcc1* knockdown (NKCC1-KD) from postnatal day 0 by removing doxycycline (Dox)-containing chow. **c** In situ hybridization images with an *Nkcc1* mRNA probe showing signals in the corpus callosum and alveus at the indicated PND in control mice (*Nkcc1*$^{tetO/tetO}$) ($n = 6$ slices from 3 animals) and NKCC1-KD mice ($n = 6$ slices from 3 animals) at PND 21 and 42. Scale bar, 100 μm. Cell counts for *Nkcc1*-positive cells. Unpaired two-sided Student's *t* test, PND 21, $t_{10} = 11.26$, $P < 0.001$; PND 42, $t_{10} = 8.72$, $P < 0.001$. Data are presented as mean ± SEM. *$P < 0.05$. **d** Changes in EPSC amplitude induced by theta burst stimulation with different numbers of bursts in control mice (10 bursts, $n = 10$; 15 bursts, $n = 6$; 20 bursts, $n = 8$, PND 18–30) and NKCC1-KD mice (10 bursts, $n = 9$; 15 bursts, $n = 8$; 20 bursts, $n = 6$, PND 19–30). Data are presented as mean ± SEM. **e** Summary histograms for the change in EPSC amplitude at 35–40 min after theta burst stimulation. Two-way ANOVA, $F_{(1,41)} = 9.80$, $P = 0.0032$. Data are presented as mean ± SEM. *$P < 0.05$. Source data are provided as a Source Data file.

$P = 0.0093$ vs WT mice), demonstrating the facilitative effect of restoring axonal plasticity on synaptic LTP (Fig. 6c, d).

*Nkcc1* overexpression in OLs in adult mice restored the plasticity of myelinated fibers and associated synaptic plasticity. To explore the behavioral significance of these changes, we restored *Nkcc1* expression in OLs in a stage-specific manner (Fig. 6e) and tested whether the induction of *Nkcc1* expression in adult mice affected learning. *Nkcc1* expression was suppressed until 5–8 weeks of age by administering a Dox chow (Dox on) and was initiated by switching from the Dox chow to a normal diet (Dox off). After 3 weeks of induction, we conducted a contextual fear conditioning test, a paradigm that assesses hippocampal-dependent spatial learning, since the neural circuits involving the distal subiculum are required for spatial working

memory[16]. Fear acquisition was measured on the conditioning day (Day 1, Fig. 6f) and contextual fear memory was measured at 1 day after training (Day 2, Fig. 6f). While there was no difference in fear acquisition between the two groups of mice (freezing time every 1 min: $F_{(1, 144)} = 0.30$, $P = 0.59$; total freezing time: $t_{27} = 0.45$, $P = 0.65$) (Fig. 6g), OL *Nkcc1*-overexpressing mice exhibited stronger freezing responses than control mice (*Nkcc1*$^{tetO/+}$) for total freezing time at 1 day after training ($t_{27} = 3.69$, $P = 0.0010$ vs control mice) (Fig. 6h). Two-way repeated ANOVA also indicated a group difference in freezing time measured every 2 min ($F_{(1, 144)} = 33.5$, $P < 0.0001$) (Fig. 6h). These results suggested that enhanced *Nkcc1*-mediated OL rejuvenation was involved in the facilitation of learning behavior.

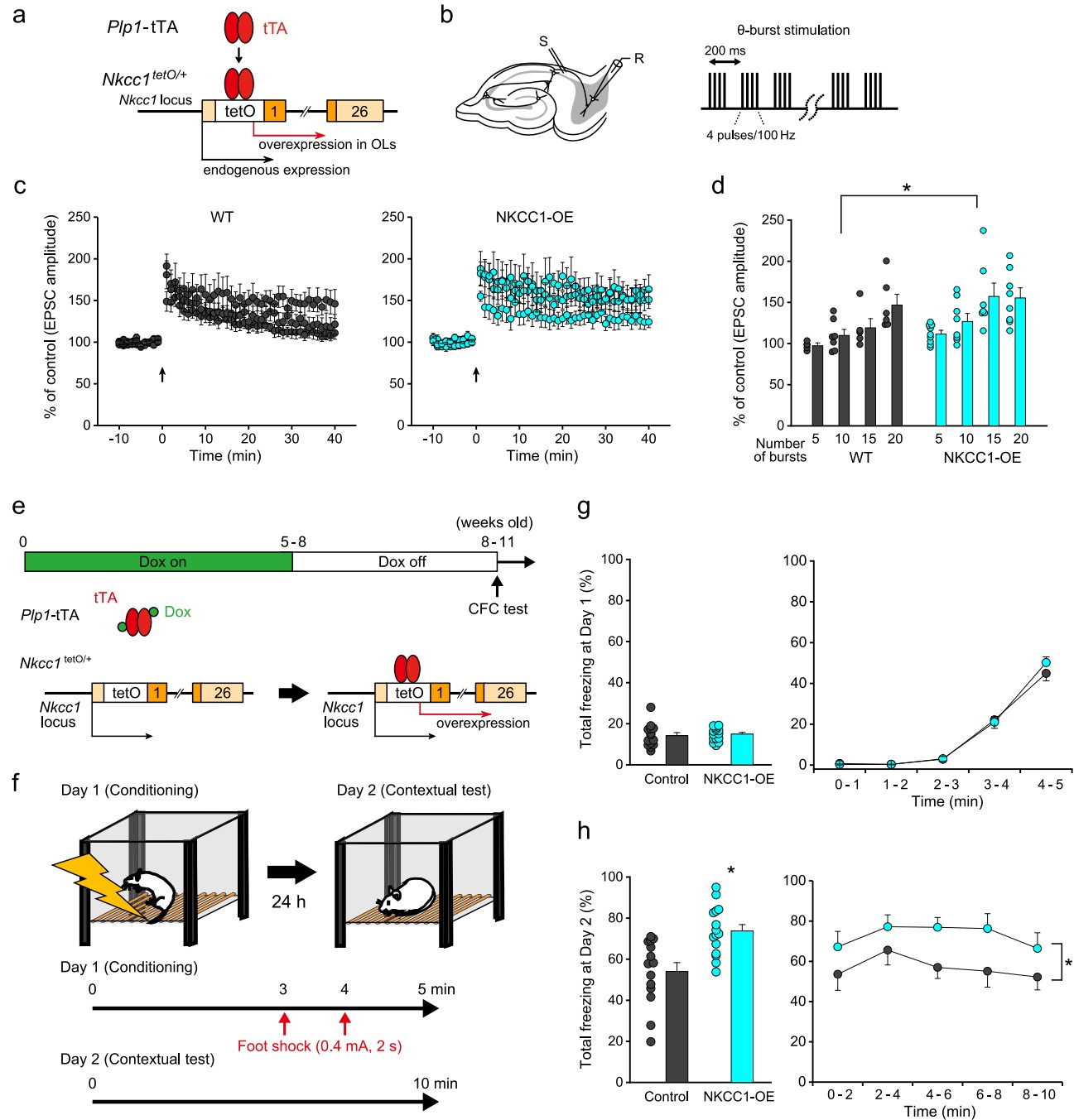

**Fig. 6 Nkcc1 overexpression in OLs enhances long-term potentiation and hippocampal-dependent learning. a** Induction of OL-specific *Nkcc1* overexpression. **b** Recording of excitatory postsynaptic currents (EPSCs) from bursting neurons in the subiculum and electrical stimulation protocol of theta burst stimulation for the induction of long-term potentiation. **c** Changes in EPSC amplitude induced by theta burst stimulation with different numbers of bursts in wild-type (WT) mice (10 bursts, $n = 7$; 15 bursts, $n = 7$; 20 bursts, $n = 6$, PND 36–43) and mice overexpressing *Nkcc1* (NKCC1-OE) in OLs (10 bursts, $n = 8$; 15 bursts, $n = 7$; 20 bursts, $n = 8$, PND 38–58). Data are presented as mean ± SEM. **d** Summary histograms for the change in EPSC amplitude at 35–40 min after theta burst stimulation. Data for 5 bursts (WT, $n = 5$; NKCC1-OE, $n = 9$) are additionally shown. Two-way ANOVA, $F_{(1,47)} = 7.17$, $P = 0.010$. Data are presented as mean ± SEM. *$P < 0.05$. **e** OL-specific *Nkcc1* overexpression in a stage-specific manner. NKCC1-OE was induced by removing doxycycline (Dox)-containing chow at 3 weeks before starting the contextual fear conditioning (CFC) test. **f** Experimental paradigm for the CFC test. **g, h** Percentage of freezing time in the CFC test (control mice, $n = 14$; NKCC1-OE mice, $n = 15$) at conditioning (Day 1) (**g**) and at contextual test (Day 2) (**h**). Unpaired two-sided Student's *t* test, $t_{27} = 3.69$, $P = 0.0010$ (left); two-way repeated-measures ANOVA, $F_{(1, 144)} = 33.5$, $P < 0.0001$ (right). Data are presented as mean ± SEM. *$P < 0.05$. Source data are provided as a Source Data file.

## Discussion

In the present study, we showed that NKCC1 on OLs is a regulator of myelinated fiber plasticity and determined the age-dependent facilitation of neural function, which was mediated by OL depolarization. Adult mice overexpressing *Nkcc1* on OLs showed larger OL-related changes in plasticity (e.g., increased conduction velocity, enhanced axonal excitation, and facilitated synaptic LTP) compared to age-matched controls, with the

magnitude of plasticity similar to that observed in young mice. The newly discovered distance from the soma-dependent increase of conduction velocity induced by OL depolarization and hippocampal-dependent learning were also related to oligodendrocyte NKCC1 activity.

The contribution of NKCC1, especially in juvenile animals, to the OL-dependent modulation of neural function was shown in the present study. In general, when cell volume is changed by osmotic loading, NKCC1 is activated to control cell volume, resulting in the movement of water across the cell membrane. Indeed, the activation of $GABA_A$ receptors on the processes of cultured OLs increases cell volume, and this change requires NKCC1 activation[17]. Therefore, it is possible that OL depolarization induced an increase in cell volume, especially in OL processes, which would lead to the axons being wrapped more tightly and increase insulation at the paranodal and intermodal regions, thereby allowing the current at one node to flow into the next node more effectively[8]. Thus, the conduction velocity of action potentials along myelinated axons is increased. In line with this theory, increased NKCC1 activity in OLs would enhance such machinery and lead to the enhancement of OL depolarization-induced axonal plasticity and vice versa. Recently, the existence of the conducting pathway formed by periaxonal and paranodal submyelin spaces was clearly proven[18], thus supporting the hypothesis that axonal conduction along myelinated axons is sensitive to morphological changes at OL processes (see also Supplementary Discussion).

Regarding the enhancement of axonal excitability, since ion channel activation is the primary contributor to axonal excitability, the changes in the properties of ion channels on axons and/or OL processes would be involved in the mechanism. The blockade of $Ba^{2+}$-sensitive $K^+$ channels inhibits the enhancement of axonal excitability induced by OL depolarization[3,19]. In addition, the threshold for the action potentials of myelinated axons is influenced by the extent of $Na^+$-channel clustering[20] and the distribution of $K^+$ channels[21] at the node, which could be modulated by structural changes in the nodal region. Since changes in node length are one form of myelin remodeling in response to neural activity[22], it is possible that the OL depolarization-induced effects on axonal excitability would be related to myelin remodeling. The magnitude of plasticity of axonal excitability was also dependent on NKCC1 activity. Thus, OLs, in addition to their prevailing role in saltatory conduction, regulate axonal conduction plasticity probably through NKCC1-mediated morphological alterations (see also Supplementary Discussion).

We should discuss the difference and relationship between myelin changes and the myelinated fiber plasticity observed in the present study. Adaptive myelination is mainly described in the context of activity-regulated structural plasticity and includes de novo myelination of previously unmyelinated or partially myelinated axons and changes in the structure of pre-existing myelin (remodeling; e.g., ion channel surface expression, changes in internode number and length, myelin thickness, myelin compaction, or node of Ranvier length)[22–24]. Conversely, myelinated fiber plasticity is a neural plastic change that emphasizes more functional aspects with actual measurements of axonal conduction, especially changes in already myelinated axons, i.e., axonal conduction plasticity in myelinated fibers. Although the ability for fully myelinated axons to show adaptive myelination is considered to be small, OL depolarization-induced neural plasticity can occur in axons that are mostly myelinated along their length by mature OLs. The timescales of adaptive myelination, i.e., the beginning of its occurrence after inducible events, required time for its establishment, and duration of significant changes are different from those of myelinated fiber plasticity. The majority of

myelin plasticity, which includes both adaptive myelination and myelin remodeling, reported requires a longer timescale (hours to days) for its appearance. Several types of myelin plasticity occur over a much longer timescale (days to weeks). The increase in myelin thickness by optogenetic stimulation of neurons or by tamoxifen-induced activation of extracellular signal-regulated kinases 1 and 2 in OLs is evaluated at 4 weeks or 5–10 days after stimulation, respectively[25,26]. Increases in myelination are observed at 7–21 days after pharmacogenetic activation of neurons or sensory enrichment[27,28]. Although myelin plasticity with a short timescale has been reported[29], these changes are observed in OL precursor cells (e.g., promotion of the cell cycle to re-enter or drive the differentiation to OLs), and neither changes in mature OLs nor a direct influence on axonal conduction have been observed. Conversely, as shown in Fig. 2c, d, myelinated fiber plasticity includes an increase in conduction velocity and an enhancement of axonal excitability, the former is early-onset, short-term plasticity (over a few minutes) and the latter occurs at several minutes after OL depolarization and reaches its peak after 20–30 min[3,4]. Therefore, in addition to the differences between the structural and functional aspects, there are considerable differences in the timescales of adaptive myelination and OL-related myelinated fiber plasticity. However, since the enhancement of axonal excitability lasts for more than 3 h[3], and since myelin sheath elongation occurs at 1–2 h after a transient increase in intracellular $Ca^{2+}$ concentration[30,31], it is possible that such $Ca^{2+}$-dependent structural changes are related to the enhancement of axonal excitability.

The subiculum is the principal target of CA1 pyramidal cells, and the neural circuits involving the subiculum play an essential role in the encoding and retrieval of long-term memory[16]. Moreover, the distal subiculum, in which bursting neurons comprise the majority of pyramidal cells, is required for spatial working memory[32]. Therefore, our results suggest that the facilitation of LTP at CA1-mid or distal subiculum synapses by *Nkcc1* overexpression in OLs accounts for the observed improvement of learning behavior. However, since *Nkcc1* is overexpressed on all *Plp1*-positive OLs, it is considered that the effects of *Nkcc1* overexpression are not specific to hippocampal function. Hence, it is important to identify other brain regions in which the effects of *Nkcc1* overexpression on OLs are clearly revealed.

In the present study, we demonstrated that OLs differentially boost axonal conduction velocity depending on their distance from the neuronal soma (Fig. 3). Since the axons enter the alveus at a distance of approximately 200 μm from the soma in the mice used in this study, the results suggest that the increase in OL depolarization-induced conduction velocity began to appear after the axons entered the alveus, and that the magnitude of this facilitative effect increased gradually. The mode value of the length of myelinating processes of alveus OLs, which corresponds to internodal length, was 72.5 μm (Supplementary Fig. 8b). Therefore, given the case of orthodromic conduction, it is suggested that, after the axon enters the alveus, the conduction velocity of the action potential is increased every time it passes each myelinated region by OL depolarization until approximately the seventh myelinated region, and that the cumulative facilitative effect peaks at approximately the same region. This heterogeneous modulation of axonal conduction suggests a more complicated form of myelinated fiber plasticity than hitherto appreciated and that several different complex operations may occur along an axon and contribute to the fine control of neural function.

Neural plasticity decreases with aging in relation to marked deficits in brain function. The rejuvenation of the brain has received much attention. Several strategies including hippocampal injection of a growth factor[33,34], intravenous

administration of blood plasma from young animals[35,36], genetically driven expansion of neural stem cells[37], and high-frequency transcranial magnetic stimulation of hippocampal-cortical brain networks[38], have been attempted to restore the age-related decline in brain function. In addition to these approaches, the targeting of OL-mediated plasticity may be a potential tool for rejuvenating the brain. Our data demonstrated that the manipulation of a single protein facilitated OL-mediated plasticity; therefore, the development of an agent that could induce or activate NKCC1 in OLs may be a key for the rejuvenation of the adult brain.

## Methods

**Animals.** All animal procedures were performed in accordance with the National Institutes of Health Guide for the Care and Use of Laboratory Animals and were approved by the Animal Research Committees of Yamagata University and Keio University. The number of animals used and their suffering were minimized. Animals were housed 2–4 per cage in the animal center of our university on a 12-h/ 12-h light/dark cycle at constant temperature (23–24 °C) and humidity (60–70%) with free access to water and rodent food.

OL-specific ChR2-expressing mice (PLP-ChR2 mice) were obtained by crossing the *Plp1*-tetracycline-controlled transcriptional activator (tTA) line with the tetracycline-controlled transcriptional activator-dependent promoter (tetO)-ChR2(C128S)-EYFP line[39–41]. OL-specific *Nkcc1*-overexpressing mice were obtained by crossing the *Plp1*-tTA line with the *Nkcc1*tetO knock-in line[15]. *Nkcc1*-overexpressing and ChR2-expressing triple transgenic mice were obtained by crossing the *Plp1*-tTA, tetO-ChR2, and *Nkcc1*tetO lines. In the slice physiology experiments, we did not use Dox to regulate gene induction timing. *Nkcc1* knockdown mice (NKCC1-KD mice, *Actin*-tTS:: *Nkcc1*tetO/tetO) were generated by using the tetracycline-inducible gene silencing system[15,42]. Homozygous *Nkcc1*tetO/tetO knock-in mice were crossed with the *Actin*-tTS line. Aged NKCC1-KD mice showed severe inner ear hypoplasia; therefore, we administered Dox (100 mg Dox kg$^{-1}$ chow; CLEA Japan, Tokyo, Japan) to pregnant mice and maintained normal NKCC1 expression during the embryonic period. We switched Dox chow to normal chow at PND 1 and started tTS-dependent knockdown (See Fig. 5b). In this Dox regimen, the developmental inner ear hypoplasia was minimized and the apparent growth retardation was not observed[15]. For the behavioral test, OL-specific *Nkcc1*-overexpressing mice were fed with Dox from conception to 3 weeks before the test, and were switched to normal chow in order to induce NKCC1 expression (see Fig. 6e). The genetic background of all transgenic mice was mixed C57BL/6 and 129SvEvTac. The PCR primer sets used for mouse genotyping were shown in Supplementary Table 1.

**Immunohistochemistry.** The mice were anesthetized deeply with ketamine (100 mg/kg) and xylazine (10 mg/kg) and perfused with a 4% paraformaldehyde phosphate-buffer solution. The brains were removed from the skull and post-fixed in the same fixative overnight. Subsequently, the brains were cryoprotected in 20% sucrose overnight, frozen, and cut at 25-µm thickness on a cryostat. The sections were mounted on silane-coated glass slides (Matsunami Glass, Osaka, Japan). The sections were incubated with the primary antibodies overnight at room temperature. The following antibodies were used: anti-PLP (1:1 dilution; rat monoclonal, clone AA3 hybridoma supernatant); anti-GFP (1:250; goat polyclonal, Rockland); and anti-NG2 (1:500; rabbit polyclonal, Millipore). The sections, then, treated with species-specific secondary antibodies conjugated to Alexa Fluor 488, 555, 594 or 647 (1:1,000; Invitrogen, Carlsbad, CA, USA) for 2 h at room temperature. Fluorescent images were obtained with an inverted microscope (BZ-X710; Keyence, Osaka, Japan) or a confocal microscope (LSM710; Carl Zeiss, Oberkochen, Germany). The list of antibodies used in this study was shown in Supplementary Table 2.

**Super-resolution microscopy analysis.** The method of a structured illumination microscope (SIM) with super-resolution microscopy (SRM) was described previously[43]. The procedure of PLP staining was described above. Prior to mounting, the concentration of 2,2'-thiodiethanol (TDE) was increased in a stepwise manner (10%, 25%, 50%, and 97% in phosphate-buffered saline [PBS]) to prevent shrinking of the samples by osmotic shock; the samples were incubated in each concentration of TDE for 5 min. Mounting medium (97% TDE, 0.24% DABCO in PBS) was applied to the samples. SIM-SRM images were obtained using a Zeiss ELYRA 3D-SIM system (Carl Zeiss, Oberkochen, Germany) equipped with an EM-CCD camera. Z-section images were obtained at an interval of 126 nm using a 64× objective lens. The number of pattern rotations of the structured illumination was adjusted to 3 in the ELYRA system. After obtaining all images, the SIM images were reconstructed.

ImageJ software (http://rsb.info.nih.gov/ij/) was used for measurements. We targeted the stratum oriens of the hippocampus. Optimal brightness and grayscale

pixel values were adjusted so as to provide the sharpest discrimination of the myelin/axon border. These traced images were converted into binary images. For the measurement of axon diameter, myelin thickness, and g-ratio (the ratio of the inner axonal diameter to the total outer diameter), 50 myelinated axons in the stratum oriens were selected randomly in each animal and examined. Axon diameter was defined as the minor axis of an ellipse-approximated axon. The median axon diameter of 50 myelinated axons was considered as the representative value of axon diameter in each animal. The g-ratio was calculated using the equation 0.5AD / (0.5AD + MT), where AD is the diameter of the axon and MT is myelin thickness. The average myelin thickness and g-ratio of 50 myelinated axons was considered as the representative value of myelin thickness and g-ratio, respectively, in each animal. For the measurement of the density of myelinated axons, the number of myelinated axons was counted in two non-overlapping fields (each field, 400–500 µm$^2$) of the stratum oriens in each animal. The number of axons in each field was divided by each area and the average from those two fields was considered as the density of myelinated axons in each animal.

**In situ hybridization.** The detailed protocol was described previously[43]. Cryosections from fixed brains were treated with proteinase K (40 µg/mL; Roche, Tokyo, Japan). After washing with PBS, the sections were acetylated with 0.25% acetic anhydride. Prehybridization was carried out for 2 h at 60 °C in hybridization buffer containing 50% formamide (Wako, Tokyo, Japan), Denhardt's solution (Nacalai Tesque, Inc., Kyoto, Japan), and 10 mg/mL salmon sperm DNA (Invitrogen, Carlsbad, CA, USA). For colorimetric in situ hybridization, the sections were incubated with hybridization buffer containing a digoxigenin (DIG)-labeled complementary RNA (cRNA) probe (*Plp1*, *Pdgfrα*, *LncOL1*, *Ctps, or Nkcc1*). After the sections were washed in buffers with serial differences in stringency, they were incubated with an alkaline phosphatase-conjugated anti-DIG antibody (1:5,000; Roche, Tokyo, Japan). The cRNA probes were visualized with freshly prepared colorimetric substrate (NBT/BCIP; Roche, Tokyo, Japan). Images of the sections were captured using an inverted light microscope (BZ-X710; Keyence).

For quantitative analysis of the expression of each RNA, cell density was calculated by dividing the number of cells expressing the target RNA in a region of interest (ROI) by the area of the ROI using ImageJ software (http://rsb.info.nih.gov/ij/). The ROI was defined as a region including the alveus, corpus callosum, and stratum oriens layer of the dorsal hippocampal CA1. We counted cell density using 8 hippocampal slices from 2 animals for each indicated PND of each RNA.

For double fluorescence in situ hybridization, the fresh frozen sections at 14-µm thickness were prepared. A FITC-labeled Nkcc1 cRNA probe and DIG-labeled *Plp1*, *Pdgfrα*, *Ctps*, or *LncOL1* cRNA probe were hybridized to the prehybridized sections. For these combinations of *Nkcc1/Plp1*, *Nkcc1/ Pdgfrα*, *Nkcc1/Ctps*, after a stringent wash, the sections were incubated with a peroxidase-conjugated anti-FITC antibody (1:1000; Roche, Tokyo, Japan) and an alkaline phosphatase-conjugated anti-DIG antibody (1:500; Roche, Tokyo, Japan). The *Nkcc1* probe was visualized with FITC using the TSA-Plus system (PerkinElmer, Waltham, MA, USA) and the probes of *Plp1*, *Pdgfrα*, and *Ctps* were visualized with alkaline phosphatase substrate Kit III (VECTOR Blue; Vector Laboratories, Burlingame, CA, USA). The fluorescence signals of VECTOR Blue were obtained by the Cy5 filter. For the combination of *Nkcc1/LncOL1*, the sections were incubated with a peroxidase-conjugated anti-FITC antibody. After quenching the peroxidase with 1% H$_2$O$_2$, the sections were incubated with a peroxidase-conjugated anti-DIG antibody (1:1000; Roche, Tokyo, Japan). *Nkcc1* and *LncOL1* probes were visualized with FITC and Cy3, respectively, using the TSA-Plus system. All sections were stained by DAPI (1 mg/mL, Sigma) for 5 min. Fluorescence images were obtained with a confocal microscope (LSM710; Zeiss).

Hybrid staining with in situ hybridization of *Plp1*, *Pdgfrα*, *LncOL1*, and *Ctps* mRNA and immunohistochemistry of PLP protein was conducted to characterize the expression stage of DM20. The in situ hybridization procedure was described above. These cRNA probes were visualized with colorimetric substrate (NBT/ BCIP). Then, we performed immunohistochemistry of PLP as described above. The sections were sequentially treated with species-specific secondary antibodies conjugated to Alexa Fluor 594 (1:1,000; Invitrogen, Carlsbad, CA, USA). The images were obtained with an all-in-one microscope (BZ-X710; Keyence).

**Slice preparation.** Hippocampal slices were prepared from 18-to 55-day-old male or female transgenic mice their wild type littermates. After the animals were decapitated under deep isoflurane anesthesia, 400-µm-thick slices were prepared at a 45° angle to the septo-temporal axis using a rotary slicer (DTY-7700; Dosaka, Kyoto, Japan) to obtain long alveus fibers in a hippocampal slice and were maintained for at least 1 h before recording at 30 °C in artificial cerebrospinal fluid (aCSF) containing (in mM) 124 NaCl, 3 KCl, 1.25 NaH$_2$PO$_4$, 2 MgSO$_4$, 2.5 CaCl$_2$, 22 NaHCO$_3$, and 10 glucose oxygenated with 95% O$_2$/5% CO$_2$.

**Electrophysiological recordings.** For recordings, a slice was transferred to a recording chamber and perfused continuously at a rate of 3 mL/min with aCSF at 30 °C. For whole-cell recordings, OLs in the alveus and pyramidal cells in the CA1 region and subiculum were visualized using an infrared differential interference

contrast microscope (E600-FN; Nikon, Tokyo, Japan) with a ×40 water immersion objective. Patch electrodes were pulled from borosilicate glass (World Precision Instruments, Sarasota, FL, USA) using a micropipette puller (P-97; Sutter Instrument Co., Novato, CA, USA). The electrodes were filled with a solution containing (in mM) 140 K-gluconate, 10 HEPES, 0.5 EGTA, 10 NaCl, 1 $MgCl_2$, 2 Mg-ATP, and 0.2 Na-GTP, adjusted to pH 7.3 with KOH. The resistance of the electrodes was set to 8–10 MΩ and 5–7 MΩ for OLs and pyramidal cells, respectively. Whole-cell recordings were performed from the soma in either voltage-clamp or current-clamp mode at near resting membrane potential (between −60 and −70 mV). Focal application of GABA (1 mM, dissolved in aCSF) was performed by pressure ejection (12 ms, 14 psi) close to the soma of the OLs (~50 μm) using a Picospritzer (General Valve, Fairfield, NJ, USA). The NKCC1 inhibitor bumetanide (10 μM) (Santa Cruz Biotechnology, Dallas, TX, USA) and $GABA_A$ receptor antagonist bicuculline (10 μM) (Wako, Osaka, Japan) were bath-applied, and the bumetanide-sensitive component was measured at 20 min after application. The resting membrane potential was determined from traces with zero-current injection or alternatively read from the built-in display after breaking into whole-cell mode (Axopatch 200B; Axon Instruments, Union City, CA, USA). To obtain input resistance, small hyperpolarizing voltage steps (10 mV, 600 ms duration) were applied using the voltage-clamp mode, and voltage was divided by the value of the current at the plateau of the response. To evaluate the magnitude of OL depolarization induced by photostimulation, the OLs were current-clamped and membrane potentials were measured in 10-s bins, except for the period from 0 to 7 s immediately after photostimulation, in which they were measured in 1-s bins, and these values were summed to give the area of OL depolarization. A short pulse (1–500 ms) of blue light (high power blue LED, 470 nm) was applied to activate the ChR2 channels expressed on the OLs. Antidromic action potentials were recorded from CA1 pyramidal cells by stimulating the axons of the recorded pyramidal cells with a fine tip bipolar tungsten stimulating electrode, and latency was measured as the time from the artifact of electrical stimulation to the start of the action potential. The start of the action potential was taken as the time when the slope of the tangent to the action potential became greater than 10 mV/ms. The action potentials induced by antidromic stimulation were recorded once every 10 or 15 s. After stable responses were obtained for more than 5 min, a blue light pulse (500 ms) was applied. The mean latency during the 3-min period immediately before the delivery of blue light was defined as the 100% level. The mean latency at 1–3 min after photostimulation was calculated to evaluate the change in axonal conduction induced by OL depolarization. The distance from the soma to each stimulating point (i.e., tip of the stimulating electrode, which was visualized as the mark of the insertion site of the stimulating electrode) was measured along the axon after staining with injected biocytin. For measuring the latency differences, the axon was stimulated at two distinct positions ($S_1$ and $S_2$) (Supplementary Fig. 5a), in which the distance from the soma was kept as constant as possible. Alternate stimuli at $S_1$ and $S_2$ were applied at 2-s intervals and repeated once every 15 s. The latency differences were divided by the linear distance between the two stimulating electrodes (ms/mm) as presented previously[44] to obtain the parameter for evaluating conduction velocity. For extracellular field recordings, the stimulating electrode was placed in the alveus of the CA1 region and CAPs were recorded from the alveus, close to the subiculum or CA2 region, using glass electrodes filled with aCSF (8–10 MΩ) in the presence of the non-NMDA glutamate receptor antagonist 6,7-dinitroquinoxaline-2,3-dione (20 μM; Sigma-Aldrich, St. Louis, MO, USA) to prevent possible contamination by synaptic responses. At the beginning of each experiment, a stimulus/response curve was established by measuring CAP amplitude, and then, the intensity of the stimulus pulse was adjusted to elicit CAPs that were 50–60% of maximal and maintained at this level throughout the experiment. Baseline responses were recorded following the delivery of the test stimuli every 15 s. After stable responses were obtained for more than 10 min, a blue light pulse (500 ms) was applied. To evaluate changes in CAP, the amplitude of CAPs was measured at 28–30 min after photostimulation and expressed as a ratio of the mean responses recorded during the 8-min period before the delivery of the light pulse. EPSCs were evoked by stimulation of the alveus between the CA1 region and subiculum, where the axons from CA1 pyramidal cells pass through to reach their target subicular neurons, using a stimulating electrode and recorded from voltage-clamped subicular neurons in the mid and distal regions of the subiculum once every 15 s. Series resistance was monitored throughout the experiments by the application of hyperpolarizing pulses through the patch pipette; if series resistance changed by more than 25%, the experiment was stopped and the data were eliminated. The amplitude of EPSCs was measured to evaluate the changes in excitatory synaptic transmission. For the LTP experiments, stimulation intensity was set to produce 40–60% of the maximum response without an action current, and theta burst stimulation (5, 10, 15, or 20 bursts, with each burst containing 4 pulses at 100 Hz and individual bursts separated by 200 ms) was delivered to induce LTP. The mean amplitude during the 10-min period immediately before the delivery of theta burst stimulation was defined as the 100% level. The mean amplitude at 35–40 min after theta burst stimulation was calculated to evaluate the magnitude of LTP. Potential and current responses were recorded using Axopatch-200B or Axopatch-1D amplifiers (Axon Instruments, Union City, CA, USA), filtered (5 kHz), and stored in a computer after conversion (digitized at 50–100 kHz) by an analog-digital converter (PCI-6023E; National Instruments, Austin, TX, USA). Data were analyzed off-line using a wave analyzing program developed by ourselves using Visual Basic (Microsoft, WA, USA) and OriginLab (Northampton, MA, USA).

**Biocytin staining.** To examine the shape of the recorded OLs, the patch electrode was filled with 0.5% biocytin. After recording was completed, the patch electrode was withdrawn carefully, and the slice was removed from the recording chamber and fixed overnight at room temperature in a solution of 4% paraformaldehyde in sodium phosphate buffer, pH 7.4. The fixed slice was rinsed in PBS (pH 7.4), treated for 30 min with 1% $H_2O_2$ to neutralize endogenous peroxidase, and then, following several rinses in PBS, incubated for 24 h at room temperature with avidin-biotinylated peroxidase complex diluted in PBS containing 0.3% Triton X-100. Finally, following a 30-min rinse in PBS, the slice was incubated with 0.06% DAB and 0.006% $H_2O_2$, diluted in PBS. The length of OL processes was measured semi-automatically using image analysis tools (Neurolucida; Mitani, Tokyo, Japan). The processes detected by the program were accepted or rejected by subsequent visual inspection based on whether their shapes and directions were generally accepted as myelinated processes.

**Contextual fear conditioning test.** The details of this experiment were reported previously[45]. The mice were trained and tested in conditioning chambers (17 × 17 × 30 cm) fitted with a stainless-steel grid floor through which footshocks could be delivered. At day 1 (conditioning test), the mice were placed in the chamber and received two electrical footshocks (2 s duration; 0.4 mA) at 3 min after the test was initiated. The mice were returned immediately to their home cage after the conditioning test. At day 2 (contextual test), the mice were placed in the same chamber for 10 min without electrical footshocks. Their behavior was recorded using a video camera placed above the chamber. Hippocampal-dependent memory was assessed by calculating the percentage of time spent freezing. Freezing behavior, which was defined as a complete lack of movement, except for respiration, was measured automatically using our in-house MATLAB code.

**Statistical analysis.** All data are expressed as the mean ± standard error of the mean. Sample size $n$ refers to the number of OLs, neurons, or hippocampal slices analyzed in electrophysiological recordings or in situ hybridization. In the behavioral test, sample size $n$ refers to the number of animals. As the initial data obtained from electrophysiological recordings were indistinguishable between male and female mice, all data were combined for statistical analysis.

For statistical analysis of cell density in in situ hybridization between the younger group (PND 21 and 28) and older group (PND 35 and 42), an unpaired two-sided Student's $t$ test was used. For statistical analysis of the membrane properties between the ≤ PND 30 and > PND 35 mice, between mice expressing ChR2 with $Nkcc1$ overexpression and mice expressing ChR2 alone, and between $Nkcc1$ knockdown mice and control mice ($Nkcc1^{tetO/tetO}$), an unpaired two-sided Student's $t$ test was used. For statistical analysis of the magnitude of OL depolarization by photostimulation between ≤ PND 30 and > PND 35 mice, and between mice expressing ChR2 with $Nkcc1$ overexpression and mice expressing ChR2 alone, two-way repeated measures ANOVA was used. For the statistical analysis of the axon diameter, myelin thickness, g-ratio, and density of myelinated axon, an unpaired two-sided Student's $t$ test was used to compare the data between the control mice and $Nkcc1$ overexpression or $Nkcc1$ knockdown mice. For the significant changes in the latency of antidromic action potentials, latency difference of antidromic action potentials, and CAP amplitude, an unpaired two-sided Student's $t$ test was used to compare the data between two groups of ChR2-expressing mice (≤PND 30 vs > PND 35, ≤PND 30 vs ≤ PND 30 in the presence of bumetanide, $Nkcc1$ overexpression with ChR2 expression vs ChR2 expression alone). Although the developmental changes continued in the examined ages, since none of the values for the magnitude of myelinated fiber plasticity were significantly different between the subdivided groups within the ≤ PND 30 or > PND 35 group, the comparison of the magnitude of myelinated fiber plasticity between the ≤ PND 30 and > PND 35 groups is appropriate. For the significant changes in magnitude and bumetanide sensitive component of GABA-induced currents, an unpaired two-sided Student's $t$ test was used to compare the data between mice expressing ChR2 with $Nkcc1$ overexpression and mice expressing ChR2 alone, and between $Nkcc1$ knockdown mice and control mice ($Nkcc1^{tetO/tetO}$). For statistical analysis of the magnitude of LTP, two-way ANOVA with Tukey's *post hoc* test, where appropriate, was used to compare the data between $Nkcc1$-overexpressing mice and wild-type control mice, and $Nkcc1$ knockdown mice and control mice ($Nkcc1^{tetO/tetO}$). Behavioral data were analyzed by an unpaired two-sided Student's $t$ test and two-way repeated-measures ANOVA with Tukey's *post hoc* test. Differences were considered statistically significant at $P < 0.05$.

**Reporting summary.** Further information on research design is available in the Nature Research Reporting Summary linked to this article.

## Data availability

All data supporting the findings of this study are provided within the paper and its Supplementary Information. A source data file is provided with this paper. All additional

information will be made available upon reasonable request to the authors. Source data are provided with this paper.

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

## Acknowledgements

The authors would like to show our greatest appreciation to Dr. Kazuhiro Ikenaka. We thank Hirotaka Nishimura for the early stage of the histological experiments. This work was supported by the Japan Society for the Promotion of Science KAKENHI Grants-in-Aid for Scientific Research (25117005 and 16K01943) and a grant-in-Aid for Transformative Research Areas (A) 'Glial Decoding' under grant number 20H05896.

## Author contributions

Y.Y., S.F., and K.F.T. designed the research; Y.Y., Y.A., and K.F.T. performed the experiments; Y.Y. and Y.A. analyzed the data; Y.Y., Y.A., and K.F.T. wrote the paper.

## Competing interests

The authors declare no competing interests.
