## [Peer Review File · Nature Communications]

Reviewers' Comments:

Reviewer #1:

Remarks to the Author:

The article of Yamazaki et al. investigates how oligodendrocyte (OL) plasticity is higher in younger (<30 days) than in older mice (>35 days), affecting neuronal conduction and long-term potentiation. The originality of this study resides on the demonstration that the expression of Na-K-Cl transporter NKCC1 is essential for this plasticity during development. To assess the role of this transporter in OL, the authors used genetic strategies to knockdown or overexpress NKCC1. Then, they employed optogenetics, electrophysiology and a behavioral approach to modulate responses in neurons of their different mouse models. Although this article shows new potential interesting aspects of the age-dependent oligodendrocyte biology and its implications in neuronal function during development and aging, the physiological role of this transporters in OL is not totally demonstrated. Several conceptual aspects should be reviewed:

1) Although Plp1 is a major myelin protein, the Plp1 gene is also expressed in oligodendrocyte precursor cells (OPCs) and premyelinating OL at postnatal stages (Mallon et al., 2002, J Neurosci). While many targeted cells in the adult Plp1-mtTA mice are probably OL, it is very likely that OPCs/premyelinating OL also constitute main targeted cells, in particular before P30 when more plasticity was observed. In fact, reporter expression in OL lineage cells was not quantitatively analyzed in this mouse at early postnatal stages (Inamura et al., 2012, Genesis) and one out of three PDGFRa-positive cells of Fig 3b in the present study seems to be positive to NKCC1 at P55 in NKCC1-OE mice. Since it is known that both OPCs and premyelinating OL express high levels of NKCC1, it is unclear how the authors excluded that the optogenetic stimulation did not activate OPCs/premyelinating OL, specially at younger stages (Fig. 2). Although the authors performed countings on the different populations of OL lineage cells in Fig. 1, they did not provide countings on OL lineage cells during development in their different mouse models, in particular those used before P30 (Chr2 and NKCC1-KD mice). What happens if Chr2-positive OPCs/pre-OL are stimulated by light? Their depolarization is probably more important than that of OL since their input resistance is higher and their resting potential is less hyperpolarized. If this is the case, the effect observed on neuronal compound action potentials (CAPs) in control and during the application of the NKCC1 blocker bumetanide could reflect an effect mediated by OPCs/premyelinating OL rather than mature OL. In fact, it is known that OPCs per se can have a number of effects on neuronal activity (see for instance LTP in Sakry et al., 2014, Plos Biol). The authors need to demonstrate by countings and physiology that they only affect OL and not OPCs or premyelinating OL in their different models/ages. Otherwise, their interpretations could be false.

2) Important conclusions of the study are based on changes observed on the latency of evoked CAPs as this parameter is considered as an indication of conduction velocity. However, the latency is subjected to many changes that are independent of conduction velocity such as the distance between electrodes on the tissue, the number of stimulated fibers in one slice compared to another and the slicing procedure. Thus, latencies do not necessarily reflect conduction velocities. The correct way to measure conduction is by recording CAPs at two distinct positions and by calculating the velocity as:

$V_c = (d_2 - d_1) / (L_2 - L_1)$, where V_c is the conduction velocity, d_1 and d_2 the distances between electrodes for each position and $L_2 - L_1$ the latency between CAP onsets for position 1 and 2. This calculation is done to rule out any uncertainty involved in the measurement of latencies. For instance, it cannot be excluded that the percentage of change of the latency during optogenetic stimulation will be largely affected by the position of the electrodes rather than conduction, particularly in younger mice. Conduction velocity should be properly measured.

3) Although context fear conditioning has been reported as a task associated with hippocampal dysfunction, other brain regions as the amygdala and the prefrontal cortex are also involved. The behavioral data in this study are poorly detailed and not very strong. First, the authors do not show the data on the acquisition phase. Is there any difference between control and mice over-expressing NKCC1, indicating differences in the amygdala rather than the hippocampus? Second, there is not a clear context extinction in neither condition (rather the behavior did not change over time in both groups). Third, the n-size is relatively small for behavioral comparisons and the reported behavioral difference between controls and mutants is weak (Fig. 5g). Further tests

should be applied to demonstrate that the hippocampus is the main region involved in these experiments. To clearly dissociate hippocampal vs non-hippocampal function during fear conditioning, animals should display a different behavior during contextual extinction, but a similar response to a conditioned stimulus (a tone cue for instance). A different protocol to test for this possibility should be done. Other tasks relevant for the hippocampus such as the T-maze spontaneous alternation task or the Morris water maze should reinforce the results.

4) It is unclear which precise ages were used in this study. The authors refer to <30 or >35 for most of the experiments. However, it would be a mistake to include experiments at P14, P20 and P30 in the same group of young mice and experiments at P35, P45 and P60 in the same group of older mice (when myelination is still ongoing and neuronal properties evolve during the first postnatal months). Ages should be homogeneous and precisely defined in the text and figure legends.

Reviewer #2:

Remarks to the Author:

amazaki et al. report novel mechanisms of how oligodendrocytes mediate functional plasticity in axonal conduction. Using transgenic mice that overexpress channelrhodopsin-2, they show that oligodendrocyte depolarization increases conduction velocity and axonal excitability in young but not adult mice. To explore why adult oligodendrocytes lose this plasticity they focus on *Nkcc1*, which is expressed in oligodendrocytes in young but not aged mice. To determine the role of *Nkcc1* they use pharmacological and genetic approaches. In their pharmacological approach, the authors find that the bumetanide-sensitive component of GABA induced inward currents is age-dependent. In the genetic approach, they find that oligodendrocyte-specific overexpression of *Nkcc1* restores oligodendrocyte-induced axonal plasticity. Furthermore, in an *Nkcc1* loss-of-function approach they find that plasticity is reduced in younger mice. Finally, they connect plasticity of axonal fibers to synaptic plasticity. In summary, the authors suggest a model in which neuronal activity triggers oligodendrocyte depolarization using *Nkcc1*-sensitive component of GABA inward currents. The depolarization in turn triggers an increase in conduction velocity and axonal excitability, which modulates synaptic plasticity. This is an interesting model of how *Nkcc1* activity may account for age-dependent oligodendrocyte-related axonal plasticity.

The main problem I have with this paper is the unclear definition of oligodendrocyte-related axonal plasticity. The authors show that oligodendrocyte increase conduction velocity and axonal excitability. However, it is not clear how this occurs. Are new myelinated segments formed? Or are existing one being remodeled? The authors need to determine whether oligodendrocyte depolarization or *Nkcc1* manipulations induces the formation of new mature oligodendrocytes and myelin sheaths. If this is not the case, the authors need to examine myelin remodeling.

A second major issue is the poorly characterized mouse models. The authors need to show whether and which cell types/stages *Nkcc1* is induced in their *Nkcc1* overexpressing transgenic mice. In their *Nkcc1* loss-of-function model it is not clear whether *Nkcc1* is only depleted in oligodendrocytes or also in other cells. There is no RNA/protein data showing that these models work. In addition, how does myelin ultrastructure look like?

In Figure 1, the distinction between NFOL and MFOL is not clear. Do *LncOL1* and *Ctps* mark different cell stages or similar?

The authors need to update their reference list. There are a large number of paper on oligodendrocyte/myelin plasticity that are not cited/discussed. Their findings need to be discussed and related to previous literature.

Reviewer #3:

Remarks to the Author:

The authors have previously published a series of studies on plasticity of CAP conduction induced

by photostimulating oligodendrocytes. The work described in this manuscript represents a significant advance over their previous studies in that the authors have identified NKCC1-mediated chloride transport as a mechanism that mediates the age-dependent plasticity. Here, they first describe their new observation that axonal conduction is enhanced more robustly by photostimulation of oligodendrocytes in younger mice than in mice older than P35. The threshold age coincides with the end of synaptogenesis and synaptic maturation, continued LTP inducibility, and myelin maintenance in the hippocampus. Pharmacological activation of NKCC1, which is more abundantly expressed in oligodendrocytes <P30, abrogates the age difference in CAP facilitation by oligodendrocyte depolarization. Overexpression of Nkcc1 in OLs in older mice enhanced CAP conduction, extended to signals arriving at longer latencies. They then show that LTP in the subiculum is diminished in young mice with tTS-induced Nkcc1 knockdown, while it is enhanced in tTA-activated Nkcc1 in older mice. Furthermore, when Nkcc1 is expressed in OLs in older mice, the mice exhibit a more exaggerated response to fear conditioning.

The manuscript is clearly written, and the data appear to be solid. The experiments are carefully designed and well controlled. The combined use of multiple sophisticated genetic manipulations is quite impressive. Conceptually this contains very exciting new data that significantly advances our understanding of the role of oligodendrocytes in neuronal plasticity. I have only very minor comments and suggestions for more effectively communicating the physiological significance to some readers who are not experts in electrophysiology.

1. On page 5, line 87, the authors provide a description of mechanisms that could trigger OL depolarization and seem to settle on NKCC1. Could they provide, perhaps in the Discussion, a more detailed rationale for why they settled on examining Nkcc1 and not the other mechanisms?

2. Along the same lines as above, could the authors provide some physiological context in which OL depolarizations could be occurring? What are some physiological events are modeled by the temporal pattern and magnitude of Chr2-mediated depolarizations used in this study?

Other minor comments

3. When they describe their results as >PNB35 or \leq PNB30 in Figure 1b (and in other figures), what were the age ranges in the groups?

4. In Figure 2, they show that Nkcc1 mRNA is present in a subpopulation of Plp1 mRNA+ oligodendrocytes. Should this be interpreted that the age-dependent decline in Nkcc1 expression is due to reduced percentage of Nkcc1+ OLs rather than reduced level of Nkcc1 expression per OL? That is, do the authors have any evidence that Nkcc1 is heterogeneously expressed among OLs, e.g. in a specific subtype described in Figure 1c, and that the relative proportion of these subtypes shifts with age? Notably, almost all of the Plp1 mRNA+ cells seem to express Nkcc1 when it is overexpressed.

5. The authors previously described a distance effect on LTP recorded from the subiculum. I am assuming that they had tried different distances between the stimulating and recording electrodes and are showing the pair with the most prominent response in Figures 4 and 5, but perhaps they could comment on that.

Overall, this is an outstanding manuscript with significant new data that is conceptually novel and exciting.

Point-by-point replies

We thank the editor and reviewers for their careful reading of our manuscript and their thoughtful comments. Their suggestions are greatly appreciated and nearly all have been incorporated into the revised manuscript. Below you will find our point-by-point responses.

The changed and additional descriptions have been highlighted in yellow in the revised manuscript.

Reviewer #1 (Remarks to the Author):

The article of Yamazaki et al. investigates how oligodendrocyte (OL) plasticity is higher in younger (<30 days) than in older mice (>35 days), affecting neuronal conduction and long-term potentiation. The originality of this study resides on the demonstration that the expression of Na-K-Cl transporter NKCC1 is essential for this plasticity during development. To assess the role of this transporter in OL, the authors used genetic strategies to knockdown or overexpress NKCC1. Then, they employed optogenetics, electrophysiology and a behavioral approach to modulate responses in neurons of their different mouse models. Although this article shows new potential interesting aspects of the age-dependent oligodendrocyte biology and its implications in neuronal function during development and aging, the physiological role of this transporters in OL is not totally demonstrated. Several conceptual aspects should be reviewed:

We appreciate that the reviewer acknowledges the quality of our study and comments to help improve our manuscript.

1) Although Plp1 is a major myelin protein, the Plp1 gene is also expressed in oligodendrocyte precursor cells (OPCs) and premyelinating OL at postnatal stages (Mallon et al., 2002, J Neurosci). While many targeted cells in the adult Plp1-mtTA mice

are probably OL, it is very likely that OPCs/premyelinating OL also constitute main targeted cells, in particular before P30 when more plasticity was observed. In fact, reporter expression in OL lineage cells was not quantitatively analyzed in this mouse at early postnatal stages (Inamura et al., 2012, Genesis) and one out of three PDGFR α -positive cells of Fig 3b in the present study seems to be positive to NKCC1 at P55 in NKCC1-OE mice. Since it is known that both OPCs and premyelinating OL express high levels of NKCC1, it is unclear how the authors excluded that the optogenetic stimulation did not activate OPCs/premyelinating OL, specially at younger stages (Fig. 2). Although the authors performed countings on the different populations of OL lineage cells in Fig. 1, they did not provide countings on OL lineage cells during development in their different mouse models, in particular those used before P30 (ChR2 and NKCC1-KD mice). What happens if ChR2-positive OPCs/pre-OL are stimulated by light? Their depolarization is probably more important than that of OL since their input resistance is higher and their resting potential is less hyperpolarized. If this is the case, the effect observed on neuronal compound action potentials (CAPs) in control and during the application of the NKCC1 blocker bumetanide could reflect an effect mediated by OPCs/premyelinating OL rather than mature OL. In fact, it is known that OPCs per se can have a number of effects on neuronal activity (see for instance LTP in Sakry et al., 2014, Plos Biol). The authors need to demonstrate by countings and physiology that they only affect OL and not OPCs or premyelinating OL in their different models/ages. Otherwise, their interpretations could be false.

In this comment, the reviewer raised two concerns:

1. The Plp1-tTA line may target OPCs and premyelinating OLs. Therefore, it is likely that ChR2 or NKCC1 was induced in these cells in our system.
2. The lack of quantitative analysis of OL stage markers in the early postnatal stage (before P30) in the model mice.

As Reviewer #1 pointed out, the 11-kb *Plp1* promoter, which was developed by Dr. Macklin, was capable of inducing EGFP expression in OPCs and premyelinating OLs (*Plp1*-EGFP transgenic line, Mallon et al., 2002). The *Plp1*-tTA transgenic line used the same promoter

(Inamura et al., 2012); therefore, tTA-mediated gene induction may occur in both stages.

We first matched the term ‘premyelinating OLs’ with the newly proposed OL stages (Fig. 1d and Marques et al., 2016). Among the new stage markers, *LncOL1* expression overlapped clearly with DM20-positive premyelinating cells (Supplementary Fig. 2a); therefore, we regarded *LncOL1* as a marker for premyelinating OLs. As a result, we exploited NG2 immunohistochemistry or *Pdgfra* *in situ* hybridization as a marker for OPCs, and DM20 immunohistochemistry (Trapp et al., 1997) or *LncOL1* *in situ* hybridization as a marker for premyelinating OLs. Using these markers, we found that Chr2-EYFP protein did not colocalize with NG2 or DM20 (Fig. 2b, Supplementary Fig. 2b), demonstrating the lack of Chr2-EYFP protein expression in OPCs and premyelinating OLs. Thus, it is unlikely that OPCs/premyelinating OLs were photostimulated in our experiments.

We also examined the induction of *Nkcc1* mRNA expression in OL-lineage cells in *Plp1-tTA::Nkcc1^{tetO}* mice. *Nkcc1* mRNA was not expressed by *Pdgfra*-positive cells (Fig. 4b), indicating the absence of *Nkcc1* induction on OPCs in *Nkcc1*-overexpressing mice. Strong *Nkcc1* mRNA signals were co-labeled with *LncOL1*, *Ctps*, or *Plp1* signals (Fig. 4b), indicating *Plp1*-tTA-mediated *Nkcc1* mRNA induction after the premyelinating stages.

We have described these results as follows and presented additional data in Figs. 2b and 4b and Supplementary Fig. 2 of the revised version of the manuscript.

Results:

Page 5, Line 90:

To reconcile this single cell RNA-sequencing-based classification with the conventional classification system (premyelinating OLs and mature OLs), we conducted double labeling with DM20, an isoform of PLP protein and a marker of premyelinating OLs (Trapp et al., 1997), and respective mRNAs. DM20 was expressed by *LncOL1*-positive cells (Supplementary Fig. 2a), indicating that newly formed OLs are equivalent to premyelinating OLs.

Page 5, Line 104:

OL precursor cells (NG2-positive) and DM20-positive premyelinating OLs (newly formed OLs) did not express Chr2 (Fig. 2b, Supplementary Fig. 2b), indicating that the OL

precursor cells/premyelinating OLs were not optogenetically manipulated, while myelin-forming and mature OLs were manipulated.

Page 10, Line 211:

Plp1-, *Ctps*-, and *LncOL1*-positive cells strongly expressed *Nkcc1* mRNA, but *Pdgfra*-positive cells did not, indicating *Nkcc1* mRNA overexpression after newly-formed OL stages (Fig. 4b).

We added quantitative data to show the number of stage-specific marker-expressing cells in WT, PLP-ChR2, or *Nkcc1* knockdown mice at PND 21 and 42 (Supplementary Figs. 3a and 10a). Our data from WT mice were consistent with the open database of the Allen Institute (data from adult mice, Supplementary Fig. 1) and data from the Marques group (P21–30 and P60 mice, Supplementary Fig. 1). In particular, in WT mice, there was no evidence that *Nkcc1* mRNA was expressed by OPCs.

In the younger stage, the population of cells expressing each marker was comparable between WT and PLP-ChR2 mice and between control and *Nkcc1* knockdown mice when interpreting the results of the electrophysiological experiments.

We have described these results as follows and presented additional data in Supplementary Figs. 1, 3a, and 10a of the revised version of the manuscript.

Results:

Page 5, Line 108:

ChR2 expression did not alter the number of OL lineage cells at PND 21 and 42 (Supplementary Fig. 3a), consistent with previous data regarding OL differentiation (Marques et al., 2016) (Supplementary Fig. 1).

Page 12, Line 268:

The number of OL lineage cells was comparable between *Nkcc1* knockdown mice and control mice at PND 21 and 42 (Supplementary Fig. 10a), except for the number of *LncOL1*-positive cells at PND 21.

References:

- Mallon, B. S. et al. Proteolipid promoter activity distinguishes two populations of NG2-positive cells throughout neonatal cortical development. *J. Neurosci.* **22**, 876–885 (2002).
- Inamura, N. et al. Gene induction in mature oligodendrocytes with a PLP-tTA mouse line. *Genesis* **50**, 424–428 (2012).
- Marques, S. et al. Oligodendrocyte heterogeneity in the mouse juvenile and adult central nervous system. *Science* **352**, 1326–1329 (2016).
- Trapp, B. D. et al. Differentiation and death of premyelinating oligodendrocytes in developing rodent brain. *J. Cell Biol.* **137**, 459–468 (1997).

2) Important conclusions of the study are based on changes observed on the latency of evoked CAPs as this parameter is considered as an indication of conduction velocity. However, the latency is subjected to many changes that are independent of conduction velocity such as the distance between electrodes on the tissue, the number of stimulated fibers in one slice compared to another and the slicing procedure. Thus, latencies do not necessarily reflect conduction velocities. The correct way to measure conduction is by recording CAPs at two distinct positions and by calculating the velocity as:

$V_c = (d_2 - d_1) / (L_2 - L_1)$, where V_c is the conduction velocity, d_1 and d_2 the distances between electrodes for each position and $L_2 - L_1$ the latency between CAP onsets for position 1 and 2. This calculation is done to rule out any uncertainty involved in the measurement of latencies. For instance, it cannot be excluded that the percentage of change of the latency during optogenetic stimulation will be largely affected by the position of the electrodes rather than conduction, particularly in younger mice. Conduction velocity should be properly measured.

To examine the changes in axonal conduction induced by OL depolarization, we recorded the antidromically conducted action potentials by whole-cell recording from CA1 pyramidal cells and CAPs by extracellular recording. We measured the latency of antidromically conducted action potentials as a parameter for evaluating the conduction velocity, not the latency of CAPs. Therefore, we consider that the influence of the recording conditions and slice preparation on the differences in the number of activated axons can be excluded.

As the reviewer pointed out, the recording of CAPs at two distinct positions and the measurements of latency differences (the difference between the two latencies; L2-L1) and the distance between two recording electrodes is an ideal method to evaluate conduction velocity. However, since the examined axons were myelinated (high conduction speed) and were not sufficiently long in the mouse hippocampal preparations, the CAP latencies were short and without variation, thus making it impossible to analyze the latency differences of CAPs accurately. The inevitably large stimulation artifact in extracellular recordings also makes it difficult to measure latency accurately. Moreover, as the reviewer mentioned, the number of axons passing through each recording electrode would be different, leading to a failure to measure the true conduction velocity.

To resolve the concern raised by the reviewer, we performed whole-cell recording from CA1 pyramidal cells, stimulated the axon at two different positions, and measured the latency differences of antidromic action potentials. To minimize the influence of various factors on the latency other than conduction velocity, we tried to keep the distance between the recording and stimulating electrodes constant (approximately 200 and 600 μm from the soma). However, due to the unknown tortuosity of the examined axons and the possible influence of unknown factors during recordings, the conduction velocity calculated by $(d_2 - d_1) / (L_2 - L_1)$ is still not accurate. To overcome this issue, we presented the values as latency differences divided by the linear distance between the two electrodes (ms/mm), as presented elsewhere (Soleng et al., 2003). Using these procedures, we could evaluate and demonstrate the change in conduction velocity as appropriately as possible.

We have described the method, results and figure legends relating to the evaluation of the conduction velocity as follows and presented these data in Supplementary Fig. 5 of the revised version of the manuscript.

Methods:

For measuring the latency differences, the axon was stimulated at two distinct positions (S_1 and S_2) (Supplementary Fig. 5a), in which the distance from the soma was kept as constant as possible. Alternate stimuli at S_1 and S_2 were applied at 2-s intervals and repeated once every 15 s. The latency differences were divided by the linear distance between the two stimulating electrodes (ms/mm) as presented previously (Soleng et al., 2003) to obtain the parameter for evaluating conduction velocity.

Results:

We validated the increase in conduction velocity and its conspicuousness in younger mice in a more rigorous way by evaluating the changes in latency differences ($t_{13} = 2.42$, $P = 0.030$ vs >PND 35) (Supplementary Fig. 5).

Figure legends for Supplementary Fig. 5:

OL-mediated plasticity of conduction velocity is significant in juvenile mice.

a, Recording of antidromic action potentials in CA1 pyramidal cells at two distinct positions and measurement of the conduction latency difference (LD; L2-L1). Scale, 2 ms, 50 mV. **b**, Electrical stimulation protocol of alternative stimulation at S_1 and S_2 to obtain LD. Time-course of the LDs (in ms/mm) of action potentials during alternative stimulation, showing that alternative stimulation itself does not affect axonal conduction velocity. **c**, Time-course of the LDs after photostimulation in PLP-ChR2 mice at >PND 35 ($n = 8$, PND 36–38), and \leq PND 30 ($n = 7$, PND 27–29). The linear distance between the two stimulating electrodes was kept as constant as possible to obtain similar LDs (0.72 ± 0.11 ms, 1.69 ± 0.20 ms/mm for >PND 35 and 0.64 ± 0.11 ms, 1.57 ± 0.18 ms/mm for \leq PND 30). **d**, Summary histogram for the changes in LDs along the axons induced by OL depolarization. $*P < 0.05$.

Reference:

Soleng, A. F. et al. Conduction latency along CA3 hippocampal axons from rat. *Hippocampus* **13**, 953-961 (2003)

3) *Although context fear conditioning has been reported as a task associated with hippocampal dysfunction, other brain regions as the amygdala and the prefrontal cortex are also involved. The behavioral data in this study are poorly detailed and not very strong. First, the authors do not show the data on the acquisition phase. Is it any difference between control and mice over-expressing NKCC1, indicating differences in the amygdala rather than the hippocampus? Second, there is not a clear context extinction in neither condition (rather the behavior did not change over time in both groups). Third, the n-size is relatively small for behavioral comparisons and the reported behavioral difference between controls and mutants is weak (Fig. 5g). Further tests should be applied to demonstrate that the hippocampus is the main region involved in these experiments. To clearly dissociate hippocampal vs non-hippocampal function during fear conditioning, animals should display a different behavior during contextual extinction, but a similar response to a conditioned stimulus (a tone cue for instance). A different protocol to test for this possibility should be done. Other tasks relevant for the hippocampus such as the T-maze spontaneous alternation task or the Morris water maze should reinforce the results.*

To address the reviewer's concerns, we measured the freezing time every minute and total freezing time on the day of the conditioning test (Day 1) to evaluate fear acquisition in the control and *Nkcc1*-overexpressing mice. Since *Nkcc1* is overexpressed on all *Plp1*-positive OLs, it is considered that the input/output functions of the amygdala are also modulated by *Nkcc1* overexpression. However, there was no difference in the acquisition phase between the two groups of mice ($P = 0.59$). We did not examine myelinated fiber plasticity in the axons towards (or output from) the amygdala. The synaptic plasticity associated with myelinated fiber plasticity in the amygdala and learning behaviors dependent solely on the amygdala were also not tested. Therefore, our data did not exclude the functional modification of the amygdala. We have described the results for fear acquisition as below and have presented the data in Fig. 6 of the revised manuscript.

We also increased the number of animals in the contextual fear conditioning tests to enhance the reliability to identify behavioral differences. The updated data are shown in Fig. 6

of the revised version.

The reviewer pointed out the absence of contextual fear extinction in both groups. We presented freezing time data only for a 10-min period on Day 2 (contextual test). Contextual fear extinction would not occur during the tested period since extinction is usually evaluated over several days.

The reviewer mentioned the need for additional experiments for hippocampus-dependent learning. As described above, the effects of *Nkcc1* overexpression may occur in other brain regions in which myelinated fibers have a role in information processing. In this study, we did not intend to demonstrate (by behavioral experiments) that the effects of *Nkcc1* overexpression were specific to hippocampal function. Therefore, we consider that our data are sufficient to support the conclusion of this study. However, as the reviewer probably thought, it would be very interesting to identify in which brain regions the effect of *Nkcc1* overexpression is clearer, and so this research is very important, but outside of the scope of the current study. We have described the limitation of our data and the possible involvement of other brain regions in the effects of *Nkcc1* overexpression in the Discussion of the revised manuscript.

Results:

After 3 weeks of induction, we conducted a contextual fear conditioning test, a paradigm that assesses hippocampal-dependent spatial learning, since the neural circuits involving the distal subiculum are required for spatial working memory¹⁶. Fear acquisition was measured on the conditioning day (Day 1, Fig. 6f) and contextual fear memory was measured at 1 day after training (Day 2, Fig. 6f). While there was no difference in fear acquisition between the two groups of mice (freezing time every 1 min: $F_{(1, 144)} = 0.30$, $P = 0.59$; total freezing time: $t_{27} = 0.45$, $P = 0.65$) (Fig. 6g), OL *Nkcc1*-overexpressing mice exhibited stronger freezing responses than control mice (*Nkcc1*^{tetO/+}) for total freezing time at 1 day after training ($t_{27} = 3.69$, $P = 0.0010$ vs control mice) (Fig. 6h). Two-way repeated ANOVA also indicated a group difference in freezing time measured every 2 min ($F_{(1, 144)} = 33.5$, $P < 0.0001$) (Fig. 6h).

Discussion:

The subiculum is the principal target of CA1 pyramidal cells, and the neural circuits

involving the subiculum play an essential role in the encoding and retrieval of long-term memory (Roy et al., 2017). Moreover, the distal subiculum, in which bursting neurons comprise the majority of pyramidal cells, is required for spatial working memory (Cembrowski et al., 2018). Therefore, our results suggest that the facilitation of LTP at CA1-mid or distal subiculum synapses by *Nkcc1* overexpression in OLs accounts for the observed improvement of learning behavior. However, since *Nkcc1* is overexpressed on all *Plp1*-positive OLs, it is considered that the effects of *Nkcc1* overexpression are not specific to hippocampal function. Hence, it is important to identify other brain regions in which the effects of *Nkcc1* overexpression on OLs are clearly revealed.

References:

Roy, D. S. et al. Distinct neural circuits for the formation and retrieval of episodic memories. *Cell* **170**, 1000-1012 (2017).

Cembrowski, M. S. et al. Dissociable structural and functional hippocampal outputs via distinct subiculum cell classes. *Cell* **173**, 1280-1292 (2018).

4) It is unclear which precise ages were used in this study. The authors refer to <30 or >35 for most of the experiments. However, it would be a mistake to include experiments at P14, P20 and P30 in the same group of young mice and experiments at P35, P45 and P60 in the same group of older mice (when myelination is still ongoing and neuronal properties evolve during the first postnatal months). Ages should be homogeneous and precisely defined in the text and figure legends.

As the reviewer pointed out, it is important to compare the age-dependent effects of OL depolarization on axonal conduction in more subdivided age groups. We have performed statistical analysis to examine whether the magnitude of myelinated fiber plasticity differs between subdivided groups within the \leq PND 30 or $>$ PND 35 group as shown below. We found that all examined values for the magnitude of myelinated fiber plasticity were not significantly

different between the subdivided groups. Therefore, although the developmental changes continue throughout the examined ages, comparing the magnitude of myelinated fiber plasticity between the \leq PND 30 and $>$ PND 35 groups is appropriate. We have described the appropriateness of grouping by PND in the Methods of the revised manuscript.

According to the reviewer's suggestion, we have shown the exact age range in the \leq PND 30 and $>$ PND 35 groups where required. To avoid difficulties with reading the text, we have shown them in the figure legends, but not in the main text.

Latency of antidromic action potentials (1–3 min after photostimulation):

\leq PND 30 group ($n = 9$): PND 18–26

PND 18–20: $93.7 \pm 0.24\%$ of baseline, $n = 4$

PND 24–26: $91.3 \pm 1.3\%$ of baseline, $n = 5$

$t_7 = 1.64$, $P = 0.14$.

\leq PND 30 (+bumetanide) group ($n = 7$): PND 20–30

PND 20–22: $96.3 \pm 0.89\%$ of baseline, $n = 3$

PND 24–30: $96.7 \pm 0.77\%$ of baseline, $n = 4$

$t_5 = 0.32$, $P = 0.76$.

$>$ PND 35 group ($n = 9$): PND 39–58

PND 39–45: $96.4 \pm 1.5\%$ of baseline, $n = 5$

PND 49–58: $96.6 \pm 1.4\%$ of baseline, $n = 4$

$t_7 = 0.083$, $P = 0.93$

Amplitude of CAP (28–30 min after photostimulation):

\leq PND 30 group ($n = 12$): PND 21–28

PND 21–22: $130.0 \pm 6.7\%$ of baseline, $n = 5$

PND 25–28: $125.9 \pm 3.2\%$ of baseline, $n = 7$

$t_{10} = 0.66$, $P = 0.54$.

\leq PND 30 (+bumetanide) group ($n = 10$): P19–28

PND 19–22: $115.2 \pm 2.7\%$ of baseline, $n = 6$
PND 24–28: $118.5 \pm 5.6\%$ of baseline, $n = 4$
 $t_8 = 0.59$, $P = 0.57$.

>PND 35 group ($n = 10$): P38–57

PND 38–44: $122.2 \pm 2.8\%$ of baseline, $n = 4$
PND 50–57: $116.5 \pm 2.4\%$ of baseline, $n = 6$
 $t_8 = 1.52$, $P = 0.17$

Latency of antidromic action potentials in *Nkcc1*-overexpressing mice (>PND 35):

Control ($n = 12$):

PND 37–44: $94.2 \pm 1.0\%$ of baseline, $n = 6$
PND 47–58: $95.9 \pm 1.2\%$ of baseline, $n = 6$
 $t_{10} = 1.19$, $P = 0.26$.

Nkcc1-overexpression ($n = 8$):

PND 37–43: $93.8 \pm 1.9\%$ of baseline, $n = 3$
PND 46–56: $90.1 \pm 1.4\%$ of baseline, $n = 5$
 $t_6 = 1.46$, $P = 0.19$.

Amplitude of CAP in *Nkcc1*-overexpressing mice (>PND 35):

Control ($n = 10$):

PND 39–44: $119.3 \pm 2.8\%$ of baseline, $n = 7$
PND 47–52: $121.5 \pm 7.4\%$ of baseline, $n = 3$
 $t_8 = 0.35$, $P = 0.74$.

Nkcc1-overexpression ($n = 8$):

PND 38–45: $134.9 \pm 3.8\%$ of baseline, $n = 4$
PND 48–60: $133.7 \pm 4.5\%$ of baseline, $n = 4$
 $t_6 = 0.21$, $P = 0.84$.

Reviewer #2 (Remarks to the Author):

Yamazaki et al. report novel mechanisms of how oligodendrocytes mediate functional plasticity in axonal conduction. Using transgenic mice that overexpress channelrhodopsin-2, they show that oligodendrocyte depolarization increases conduction velocity and axonal excitability in young but not adult mice. To explore why adult oligodendrocytes lose this plasticity they focus on Nkcc1, which is expressed in oligodendrocytes in young but not aged mice. To determine the role of Nkcc1 they use pharmacological and genetic approaches. In their pharmacological approach, the authors find that the bumetanide-sensitive component of GABA induced inward currents is age-dependent. In the genetic approach, they find that oligodendrocyte-specific overexpression of Nkcc1 restores oligodendrocyte-induced axonal plasticity. Furthermore, in an Nkcc1 loss-of-function approach they find that plasticity is reduced in younger mice. Finally, they connect plasticity of axonal fibers to synaptic plasticity. In summary, the authors suggest a model in which neuronal activity triggers oligodendrocyte depolarization using Nkcc1-sensitive component of GABA inward currents. The depolarization in turn triggers an increase in conduction velocity and axonal excitability, which modulates synaptic plasticity. This is an interesting model of how Nkcc1 activity may account for age-dependent oligodendrocyte-related axonal plasticity.

We are delighted that this reviewer states that “This is an interesting model of how Nkcc1 activity may account for age-dependent oligodendrocyte-related axonal plasticity”.

1) The main problem I have with this paper is the unclear definition of oligodendrocyte-related axonal plasticity. The authors show that oligodendrocyte increase conduction velocity and axonal excitability. However, it is not clear how this occurs. Are new myelinated segments formed? Or are existing one being remodeled? The authors need to determine whether oligodendrocyte depolarization or Nkcc1 manipulations induces the

formation of new mature oligodendrocytes and myelin sheaths. If this is not the case, the authors need to examine myelin remodeling.

4) The authors need to update their reference list. There are a large number of paper on oligodendrocyte/myelin plasticity that are not cited/discussed. Their findings need to be discussed and related to previous literature.

As the definition and mechanisms of OL-related myelinated fiber plasticity examined in the present study are closely related to those of myelin plasticity, we reply to these comments together.

As is well known, myelin plasticity includes activity-regulated *de novo* myelination of previously unmyelinated or partially myelinated axons and changes in the structure of pre-existing myelin (remodeling) (Bonetto et al., 2020; Chapman and Hill, 2020; Monje, 2018). Conversely, we use the term ‘myelinated fiber plasticity’ to describe plastic changes that emphasize more functional aspects with actual measurements of axonal conduction. More specifically, it is axonal conduction plasticity in already myelinated fibers. We describe the definition of OL-related myelinated fiber plasticity and the difference and relationship between myelin plasticity and myelinated fiber plasticity (i.e., how myelinated fiber plasticity is incorporated into myelin plasticity) in the Discussion of the revised manuscript as follows. The relationship between these different types of plasticity is also mentioned in the mechanism section of the Discussion as shown below.

We should discuss the difference and relationship between so-called myelin plasticity and the myelinated fiber plasticity observed in the present study. Myelin plasticity is mainly described in the context of activity-regulated structural plasticity and includes *de novo* myelination of previously unmyelinated or partially myelinated axons and changes in the structure of pre-existing myelin (remodeling; e.g., ion channel surface expression, changes in internode number and length, myelin thickness, myelin compaction, or node of Ranvier length) (Bonetto et al., 2020; Chapman and Hill, 2020; Monje, 2018). Conversely, myelinated fiber plasticity is a neural plastic change that emphasizes more functional aspects with actual measurements of axonal conduction, especially changes in already

myelinated axons, i.e., axonal conduction plasticity in myelinated fibers. Although the ability for fully myelinated axons to show myelin plasticity is considered to be small, OL depolarization-induced plasticity can occur in axons that are mostly myelinated along their length by mature OLs. The timescales of myelin plasticity, i.e., the beginning of its occurrence after inducible events, required time for its establishment, and duration of significant changes are different from those of myelinated fiber plasticity. The majority of myelin plasticity reported requires a longer timescale (hours to days) for its appearance. Several types of myelin plasticity occur over a much longer timescale (days to weeks). The increase in myelin thickness by optogenetic stimulation of neurons or by tamoxifen-induced activation of extracellular signal-regulated kinases 1 and 2 in OLs is evaluated at 4 weeks or 5–10 days after stimulation, respectively (Gibson et al., 2014; Jeffries et al., 2016). Increases in myelination are observed at 7–21 days after pharmacogenetic activation of neurons or sensory enrichment (Mitew et al., 2018; Hughes et al., 2018). Although myelin plasticity with a short timescale has been reported (Almeida and Lyons, 2017), these changes are observed in OL precursor cells (e.g., promotion of the cell cycle to re-enter or drive the differentiation to OLs), and neither changes in mature OLs nor a direct influence on axonal conduction have been observed. Conversely, as shown in Fig. 2c, d, myelinated fiber plasticity includes an increase in conduction velocity and an enhancement of axonal excitability, the former is early-onset, short-term plasticity (over a few minutes) and the latter occurs at several minutes after OL depolarization and reaches its peak after 20–30 min (Yamazaki et al., 2014; 2019a). Therefore, in addition to the differences between the structural and functional aspects, there are considerable differences in the timescales of myelin plasticity and OL-related myelinated fiber plasticity. However, since the enhancement of axonal excitability lasts for more than 3 h (Yamazaki et al., 2014), and since myelin sheath elongation occurs at 1–2 h after a transient increase in intracellular Ca^{2+} concentration (Baraban et al., 2018; Krasnow et al., 2018), it is possible that such Ca^{2+} -dependent structural changes are related to the enhancement of axonal excitability.

This reviewer also asked about the mechanism underlying myelinated fiber plasticity. Combined with the findings of this study, we have now described the possible mechanism in detail in

response to the reviewer's request. However, since the required descriptions about the mechanism contain previously discussed matters, we feel it is inappropriate to include all of them in the main text. Therefore, we have described the essential part in the main text and further details are included in the independent section of the Supplementary Discussion as follows.

For the main text:

It is possible that OL depolarization induced an increase in cell volume, especially in OL processes, which would lead to the axons being wrapped more tightly and increase insulation at the paranodal and intermodal regions, thereby allowing the current at one node to flow into the next node more effectively (Yamazaki et al., 2007; 2010). Thus, the conduction velocity of action potentials along myelinated axons is increased. In line with this theory, increased NKCC1 activity in OLs would enhance such machinery and lead to the enhancement of OL depolarization-induced axonal plasticity and *vice versa*. Recently, the existence of the conducting pathway formed by periaxonal and paranodal submyelin spaces was clearly proven (Cohen et al., 2020), thus supporting the hypothesis that axonal conduction along myelinated axons is sensitive to morphological changes at OL processes (see also Supplementary Discussion).

Regarding the enhancement of axonal excitability, since ion channel activation is the primary contributor to axonal excitability, the changes in the properties of ion channels on axons and/or OL processes would be involved in the mechanism. The blockade of Ba²⁺-sensitive K⁺ channels inhibits the enhancement of axonal excitability induced by OL depolarization (Yamazaki et al., 2014; 2019b). In addition, the threshold for the action potentials of myelinated axons is influenced by the extent of Na⁺-channel clustering (Eshed-Eisenbach and Peles, 2019) and the distribution of K⁺ channels (Battefeld et al., 2014) at the node, which could be modulated by structural changes in the nodal region. Since changes in node length are one form of myelin remodeling in response to neural activity (Chapman and Hill, 2020), it is possible that the OL depolarization-induced effects on axonal excitability would be related to myelin remodeling. The magnitude of plasticity of axonal excitability was also dependent on NKCC1 activity. Thus, OLs, in addition to their prevailing role in saltatory conduction, regulate axonal conduction plasticity probably

through NKCC1-mediated morphological alterations (see also Supplementary Discussion).

For the Supplementary Discussion:

Myelinated fiber plasticity, as described in this study, includes an increase in conduction velocity and an enhancement of axonal excitability. Conduction velocity along myelinated axons is influenced by axonal (diameter and axoplasmic conductance), nodal (area, capacitance, and conductance), and internodal (internodal length and myelin capacitance) parameters (Ritchie, 1995). The increase in conduction velocity induced by OL depolarization is very likely due to a morphological change in OL processes.

Since Chr2 is non-selective cation channel, the activation of Chr2 and subsequent activation of voltage-dependent ion channels induce cation influx. In physiological conditions, the activation of non-NMDA and NMDA receptors contributes to ion influx with depolarization by responding to glutamate released from neurons. As the intracellular volume of the myelinating processes is very small, ion influx across the plasma membrane leads to structural changes in the processes of OLs as a result of osmotic swelling. Structural changes in the myelinating processes influence the extent of insulation at the paranodal and internodal regions. As the periaxonal and paranodal submyelin spaces form a current pathway (Cohen et al., 2020), axonal conduction along myelinated axons is sensitive to morphological changes at OL processes. When the paranodal loops swell, they may wrap the axons more tightly and the insulation at the paranodal region may increase, resulting in more effective current flow from one node to the next (Yamazaki et al., 2010). Thus, the conduction velocity of action potentials would be increased. The contribution of NKCC1 activity to the increase in conduction velocity induced by OL depolarization strongly supports the notion that the morphological change by osmotic alteration is a probable mechanism for myelinated fiber plasticity. Since the increase in conduction velocity appears within a few minutes after OL depolarization and is sustained for approximately 10 min, it is unlikely that *de novo* myelination and myelin remodeling are involved in the plastic changes of conduction velocity.

Since OLs are depolarized in response to extracellular K^+ concentrations increased by neuronal activity, the involvement of K^+ channels in the observed axonal plasticity is suspected. Ba^{2+} and 4-AP (which act on different K^+ channels) both inhibit the plastic

changes related to the increase in conduction velocity (Yamazaki et al., 2014; 2019b). While 4-AP has no significant effect on light-evoked OL depolarization, Ba^{2+} significantly suppresses depolarization. Thus, the changes in 4-AP-sensitive K^+ channels expressed on axons are likely involved in the increase in conduction velocity. The suppressive effect of Ba^{2+} is probably due to the suppression of OL depolarization by photostimulation.

In the plasticity of axonal excitability, the amplitude and area of CAP are increased, while CAP width is unchanged. These changes in CAP parameters result from the increase in the number of firing axons by electrical stimulation and/or an increase in the amplitude and/or duration of each action potential conducted along each axon, but not from an increase in conduction velocity (Yamazaki et al., 2014). However, through analysis of the paired-pulse ratio at destination synapses, the changes in the kinetics of each action potential can be excluded (Yamazaki et al., 2019a). Thus, the increase in axon excitability is due to a decrease in the threshold for the generation of action potentials.

It is clear that ion channel activation for action potential generation and the maintenance of the local ionic environment to allow for action potential propagation mainly contribute to axonal excitability (Kiernan and Kaji, 2013). The enhancement of axonal excitability is inhibited by the application of Ba^{2+} , but not by 4-AP. Since Ba^{2+} also significantly suppresses light-evoked OL depolarization, it is suggested that the inhibition of OL depolarization caused by Ba^{2+} application inhibits the enhancement of axonal excitability, and that the changes in the properties of Ba^{2+} -sensitive K^+ channels on axons and/or OLs (as a result of OL depolarization) lead to the enhancement of axonal excitability (Yamazaki et al., 2014).

In addition to the contribution of K^+ channels, since the plasticity of axonal excitability occurs over a slower and longer timescale (beginning at several minutes after OL depolarization and lasting for more than 3 h), it could be related to a certain type of myelin remodeling. In myelinated axons, the extent of Na^+ channel clustering (Eshed-Eisenbach and Peles, 2019) and the distribution of K^+ channels (Battefeld et al., 2014) at the node affect the action potential threshold of myelinated axons. Therefore, the expression location and extent of clustering of ion channels at the node (which are affected by changes in nodal structure) would be involved in the plasticity of axonal excitability. The change in node length (Chapman and Hill, 2020) or Ca^{2+} -dependent elongation of OL

processes occurring within 1–2 h (Baraban et al., 2018; Krasnow et al., 2018), both of which are one form of myelin remodeling in response to neural activity, would be related to the OL depolarization-induced enhancement of axonal excitability.

Related to the mechanisms for increased conduction velocity and enhanced axonal excitability, we must mention the roles of neurotransmitters, since it is possible that OLs signal to axons using neuroactive substances and since axonal conduction is regulated by neurotransmitter-mediated mechanisms. Similarly, as OL depolarization decreases K^+ buffering, it is possible that the extracellular K^+ concentration around axons increases, resulting in increased conduction velocity and enhanced axonal excitability. However, pharmacological experiments applying various neurotransmitter receptor antagonists have not shown significant effects on axonal conduction (Yamazaki et al., 2014). Thus, it is unlikely that the release of neuroactive substances from depolarizing OLs is involved in myelinated fiber plasticity. Moreover, it has been confirmed that a transient increase in extracellular K^+ concentration itself does not contribute to myelinated fiber plasticity (Yamazaki et al., 2014).

The magnitude of the plasticity of axonal excitability also varies depending on NKCC1 activity. Thus, OLs (in addition to their prevailing role in saltatory conduction) regulate axonal conduction plasticity in conjunction with changes in K^+ channels, probably through NKCC1-mediated morphological alterations which could be integrated into myelin plasticity. It is possible that structural myelin plasticity occludes the changes in the properties of ion channels.

References:

Bonetto, G., Kamen, Y., Evans, K. A. & Káradóttir, R. T. Unraveling myelin plasticity.

Front. Cell. Neurosci. **14**, 156 (2020)

Chapman, T.W. & Hill, R. A. Myelin plasticity in adulthood and aging. *Neurosci. Lett.* **715**, 134645 (2020)

Monje, M. Myelin plasticity and nervous system function. *Annu. Rev. Neurosci.* **41**, 61-76 (2018)

- Gibson, E. M., et al. Neuronal activity promotes oligodendrogenesis and adaptive myelination in the mammalian brain. *Science* **344**, 1252304 (2014)
- Jeffries, M. A. et al. ERK1/2 Activation in preexisting oligodendrocytes of adult mice drives new myelin synthesis and enhanced CNS function. *J. Neurosci.* **36**, 9186-200 (2016)
- Mitew, S. et al. Pharmacogenetic stimulation of neuronal activity increases myelination in an axon-specific manner. *Nat. Commun.* **9**, 306 (2018)
- Hughes, E. G. et al. Myelin remodeling through experience-dependent oligodendrogenesis in the adult somatosensory cortex. *Nat. Neurosci.* **21**, 696–706 (2018)
- Almeida, R. G. & Lyons, D. A. On myelinated axon plasticity and neuronal circuit formation and function. *J. Neurosci.* **37**, 10023–10034 (2017)
- Yamazaki, Y. et al. Short- and long-term functional plasticity of white matter induced by oligodendrocyte depolarization in the hippocampus. *Glia* **62**, 1299–1312 (2014)
- Yamazaki, Y. et al. Region- and cell type-specific facilitation of synaptic function at destination synapses induced by oligodendrocyte depolarization. *J. Neurosci.* **39**, 4036-4050 (2019a)
- Baraban, M., Koudelka, S. & Lyons, D.A. Ca²⁺ activity signatures of myelin sheath formation and growth in vivo. *Nat. Neurosci* **21**, 19–23 (2018)
- Krasnow, A. M et al. Regulation of developing myelin sheath elongation by oligodendrocyte calcium transients in vivo. *Nat. Neurosci.* **21**, 24-28 (2018)
- Yamazaki, Y. et al. Modulatory effects of oligodendrocytes on the conduction velocity of action potentials along axons in the alveus of the rat hippocampal CA1 region. *Neuron Glia*

Biol. **3**, 325–334 (2007)

Yamazaki, Y. et al. Oligodendrocytes: facilitating axonal conduction by more than myelination. *Neuroscientist* **16**, 11–18 (2010).

Cohen, C. C. H. et al. Saltatory conduction along myelinated axons involves a periaxonal nanocircuit. *Cell* **180**, 311–322 (2020)

Yamazaki, Y. Oligodendrocyte physiology modulating axonal excitability and nerve conduction. *Adv. Exp. Med. Biol.* **1190**, 123–144 (2019b)

Eshed-Eisenbach, Y. & Peles, E. The clustering of voltage-gated sodium channels in various excitable membranes. *Dev. Neurobiol.* doi: 10.1002/dneu.22728 (2019)

Battefeld, A. et al. Heteromeric Kv7.2/7.3 channels differentially regulate action potential initiation and conduction in neocortical myelinated axons. *J. Neurosci.* **34**, 3719–3732 (2014)

Ritchie JM. Physiology of axon. In: Waxman SG, Kocsis JD, Stys PK editors. The axon. New York: Oxford UP. pp 68–96 (1995)

Kiernan, M. C. & Kaji, R. Physiology and pathophysiology of myelinated nerve fibers. In: Said G, Krarup C (eds) Peripheral nerve disorders. Handbook of clinical neurology, vol 115. Elsevier, Amsterdam, pp 43–53 (2013)

2) *A second major issue is the poorly characterized mouse models. The authors need to show whether and which cell types/stages Nkcc1 is induced in their Nkcc1 overexpressing transgenic mice. In their Nkcc1 loss-of-function model it is not clear whether Nkcc1 is only depleted in oligodendrocytes or also in other cells. There is no RNA/protein data showing*

that these models work. In addition, how does myelin ultrastructure look like?

We performed double fluorescence *in situ* hybridization in *Nkcc1*-overexpressing mice at PND 42 and found that *Nkcc1* mRNA was expressed in *Plp1*-, *Ctps*-, and *LncOL1*-positive cells, but not in *Pdgfra*-positive cells. We have presented these data in Fig. 5b in the revised manuscript.

We used *Actin*-tTS to knock down the gene of interest and all cells were targeted by using the tTS-tetO system. Thus, *Nkcc1* was depleted in all cells including neurons and OL-lineage cells. We have presented the *Nkcc1* mRNA *in situ* hybridization data in Fig. 5c.

To examine the ultrastructure of myelin, we performed super-resolution microscopy analysis in *Nkcc1*-overexpressing mice at PND 42 and *Nkcc1*-knockdown mice at PND 21 and 42. Although the spatial resolution of structured illumination microscopy is approximately 100 nm, we previously demonstrated that myelin thickness and axon diameter can be determined by this method (Abe et al., 2019). Axon diameter, myelin thickness, and g-ratio in *Nkcc1*-overexpressing and *Nkcc1*-knockdown mice were comparable to those in age-matched control mice. We have shown these data in Supplementary Fig. 7b–d and Fig. 10b–d in the revised manuscript.

Reference:

Abe, Y. et al. Correlative study using structural MRI and super-resolution microscopy to detect structural alterations induced by long-term optogenetic stimulation of striatal medium spiny neurons. *Neurochem. Int.* **125**, 163–174 (2019).

3) In Figure 1, the distinction between NFOL and MFOL is not clear. Do LncOL1 and Ctps mark different cell stages or similar?

We think that *LncOL1* and *Ctps* mark different cell stages because some of the *LncOL1*-expressing cells also expressed DM20, but none of the *Ctps*-expressing cells did. However, there was an overlap between them, which is shown in single cell RNA-sequencing data from the Allen Institute (upper in Supplementary Fig. 1) and Marques et al. (2016) (lower)

as shown below. We revised Fig. 1d accordingly.

References:

Allen Brain Map: Cell Types Database: RNA-Seq Data: Mouse.

https://celltypes.brain-map.org/rnaseq/mouse_ctx-hip_10x.

Marques, S. et al. Oligodendrocyte heterogeneity in the mouse juvenile and adult central nervous system. *Science* **352**, 1326–1329 (2016). Database of Marques et al., 2016.

<http://linnarssonlab.org/oligodendrocytes/>.

Reviewer #3 (Remarks to the Author):

The authors have previously published a series of studies on plasticity of CAP conduction induced by photostimulating oligodendrocytes. The work described in this manuscript represents a significant advance over their previous studies in that the authors have identified NKCC1-mediated chloride transport as a mechanism that mediates the age-dependent plasticity. Here, they first describe their new observation that axonal conduction is enhanced more robustly by photostimulation of oligodendrocytes in younger mice than in mice older than P35. The threshold age coincides with the end of synaptogenesis and synaptic maturation, continued LTP inducibility, and myelin maintenance in the hippocampus. Pharmacological activation of NKCC1, which is more abundantly expressed in oligodendrocytes <P30, abrogates the age difference in CAP facilitation by oligodendrocyte depolarization. Overexpression of Nkcc1 in OLs in older mice enhanced CAP conduction, extended to signals arriving at longer latencies. They then show that LTP in the subiculum is diminished in young mice with tTS-induced Nkcc1 knockdown, while it is enhanced in tTA-activated Nkcc1 in older mice. Furthermore, when Nkcc1 is expressed in OLs in older mice, the mice exhibit a more exaggerated response to fear conditioning.

The manuscript is clearly written, and the data appear to be solid. The experiments are carefully designed and well controlled. The combined use of multiple sophisticated genetic manipulations is quite impressive. Conceptually this contains very exciting new data that significantly advances our understanding of the role of oligodendrocytes in neuronal plasticity. I have only very minor comments and suggestions for more effectively communicating the physiological significance to some readers who are not experts in electrophysiology.

We are delighted that this reviewer states that “Conceptually this contains very exciting new data that significantly advances our understanding of the role of oligodendrocytes in neuronal plasticity”.

1) On page 5, line 87, the authors provide a description of mechanisms that could trigger OL depolarization and seem to settle on NKCC1. Could they provide, perhaps in the Discussion, a more detailed rationale for why they settled on examining *Nkcc1* and not the other mechanisms?

As suggested by Reviewer #3, we have now provided the reason why we focused on NKCC1 activity for the mechanism of OL depolarization-induced myelinated fiber plasticity in the Introduction of the revised manuscript, as shown below.

Page 3, Line 57:

OLs and myelin show morphological changes in response to neural activity, and such morphological dynamics are related to changes in axonal conduction (Huff et al., 2011).. Since the intracellular volume of the myelinating processes is very small, neural activity-induced OL depolarization technically leads to an increase in the volume of myelinating processes as a result of osmotic swelling (Káradóttir et al., 2005). The application of GABA to OLs induces depolarizing responses (Yamazaki et al., 2010) due to the presence of Cl⁻ transporters (Hoppe & Kettenmann, 1989). Furthermore, Na⁺-K⁺-Cl⁻ co-transporter 1 (NKCC1) is a cardinal molecule for cell volume regulation. Therefore, oligodendrocytic NKCC1 expression may underlie myelinated fiber plasticity.

References:

Huff, T. B. et al. Real-time CARS imaging reveals a calpain-dependent pathway for paranodal myelin retraction during high-frequency stimulation. *PLoS One* **6**, e17176 (2011).

Káradóttir, R., Cavelier, P., Bergersen, L. H. & Attwell, D. NMDA receptors are expressed in oligodendrocytes and activated in ischaemia. *Nature* **438**, 1162–1166 (2005).

Yamazaki, Y. et al. Oligodendrocytes: facilitating axonal conduction by more than myelination. *Neuroscientist* **16**, 11–18 (2010).

Hoppe, D. & Kettenmann, H. Carrier-mediated Cl⁻ transport in cultured mouse oligodendrocytes. *J. Neurosci. Res.* **23**, 467–475 (1989).

2) Along the same lines as above, could the authors provide some physiological context in which OL depolarizations could be occurring? What are some physiological events are modeled by the temporal pattern and magnitude of ChR2-mediated depolarizations used in this study?

As suggested by the reviewer, we have now described the physiological context for OL depolarization, as shown below.

Page 6, Line 110:

Optical stimulation mimicked the physiological relevant depolarization of OLs, i.e., the magnitude of depolarization by ChR2 activation was comparable to that induced by theta rhythm electrical stimulation, which reflects physiological activity in the hippocampus.

3) When they describe their results as >PNB35 or ≤PNB30 in Figure 1b (and in other figures), what were the age ranges in the groups?

According to the reviewer's suggestion, we have shown the exact age ranges in the ≤PND 30 and >PND 35 groups in the figure legends where needed.

4) In Figure 2, they show that Nkcc1 mRNA is present in a subpopulation of Plp1 mRNA+ oligodendrocytes. Should this be interpreted that the age-dependent decline in Nkcc1 expression is due to reduced percentage of Nkcc1+ OLs rather than reduced level of Nkcc1

expression per OL? That is, do the authors have any evidence that Nkcc1 is heterogeneously expressed among OLs, e.g. in a specific subtype described in Figure 1c, and that the relative proportion of these subtypes shifts with age? Notably, almost all of the Plp1 mRNA+ cells seem to express Nkcc1 when it is overexpressed.

Regarding naive *Nkcc1* mRNA expression, please see the tSNE map for *Nkcc1* (Supplementary Fig. 1, Marques et al., 2016). The sample data were collected from PND 21 and 60 mouse brains; thereby, young and adult OL data are mixed. In this condition, *Nkcc1* mRNA expression seems to be observed in the whole *Plp1*-positive population. On the basis of our *Nkcc1* colorimetric *in situ* hybridization data at PND 21 and 42, we are sure of the significant reduction in the number of *Nkcc1*-expressing (detected) cells over time (Fig. 2e), which coincided with the significant reduction in the number of young OL marker-expressing cells (Fig. 1e). However, we are not confident that our double fluorescence *in situ* hybridization approach can be used to determine the percentage of *Nkcc1*-positive cells for each population marker. Thus, it is difficult for us to address whether the age-dependent decline of *Nkcc1* expression was due to the reduced percentage of *Nkcc1*-positive OLs or the reduced level of *Nkcc1* expression per OL.

According to the measurement of bumetanide-sensitive GABA currents, all of the OLs we recorded possessed a bumetanide-sensitive GABA current regardless of their magnitude, suggesting that the entire *Plp1*-positive population seems to express *Nkcc1* and the level of *Nkcc1* expression per cell seems to decline over time.

We added double fluorescence *in situ* hybridization images to show the *Plp1*-positive population including the *LncOLI*- and *Ctps*-positive populations with *Nkcc1* mRNA expression induced by *Plp1*-tTA. Regarding *Nkcc1* overexpression, we added a paragraph to explain the *Nkcc1* overexpression pattern as shown below. We cannot fully address Reviewer #3's question; however, our description in the main text faithfully explains the pattern.

Page 10, Line 208:

We crossed the *Nkcc1*^{tetO} knock-in line (Watabe et al., 2017) with the *Plp1*-tTA line to establish tTA-dependent *Nkcc1* induction. *In situ* hybridization showed the overexpression of *Nkcc1* mRNA in older mice (>PND 42) compared with age-matched control mice (*Nkcc1*^{tetO/+}) (Fig. 4a). *Plp1*-, *Ctps*-, and *LncOLI*-positive cells strongly expressed *Nkcc1*

mRNA, but *Pdgfra*-positive cells did not, indicating *Nkcc1* mRNA overexpression after newly-formed OL stages (Fig. 4b).

Reference:

Watabe, T. et al. Time-controllable *Nkcc1* knockdown replicates reversible hearing loss in postnatal mice. *Sci. Rep.* **7**, 13605 (2017).

5) The authors previously described a distance effect on LTP recorded from the subiculum. I am assuming that they had tried different distances between the stimulating and recording electrodes and are showing the pair with the most prominent response in Figures 4 and 5, but perhaps they could comment on that.

To investigate the effects of NKCC1 activity on the induction of LTP, we recorded EPSCs from bursting cells at the mid and distal regions of the subiculum. As the reviewer pointed out, since the effects of OL depolarization on axonal conduction differed according to axonal position, it is possible that the effects on synaptic function at the destination synapses vary depending on the stimulation site. To minimize this possible influence, we made the linear distance between the stimulating and recording electrodes constant (approximately 600 μm). We have added a description of this to the revised manuscript.

Overall, this is an outstanding manuscript with significant new data that is conceptually novel and exciting.

We are very grateful for this compliment.

Reviewers' Comments:

Reviewer #1:

Remarks to the Author:

Overall, the authors have properly addressed my concerns with the addition of new data and discussion of their results. Their manuscript constitute an advanced on how a Nkcc1-dependent oligodendrocyte mechanism influences neuronal function in hippocampal circuits, in particular during early age when the system is more plastic.

Reviewer #2:

Remarks to the Author:

The authors have answered all of my comments

Reviewer #3:

Remarks to the Author:

The authors have adequately addressed all of my comments on the previous version of the manuscript. Even though I was not the reviewer who requested it, I appreciate the additional discussion on myelinated fiber plasticity and myelin plasticity. It reinforces the novelty and significance of the authors findings. I have no further comments.

Point-by-point replies

Reviewer #1 (Remarks to the Author):

Overall, the authors have properly addressed my concerns with the addition of new data and discussion of their results. Their manuscript constitute an advanced on how an Nkcc1-dependent oligodendrocyte mechanism influences neuronal function in hippocampal circuits, in particular during early age when the system is more plastic.

We thank the reviewer for the insightful feedback during the review process and the positive comments.

Reviewer #2 (Remarks to the Author):

The authors have answered all of my comments.

No other issue was raised by the reviewer on the revised manuscript. We thank the reviewer for the constructive remarks and suggestions during the review process.

Reviewer #3 (Remarks to the Author):

The authors have adequately addressed all of my comments on the previous version of the manuscript. Even though I was not the reviewer who requested it, I appreciate the additional discussion on myelinated fiber plasticity and myelin plasticity. It reinforces the novelty and significance of the authors' findings. I have no further comments.

We are happy to hear that we have addressed all previous concerns satisfactorily and thank the reviewer for supporting our manuscript.